# Nrg1 intracellular signaling regulates the development of interhemispheric callosal axons in mice

Ángela Rodríguez-Prieto[1] ⓘ, Isabel Mateos-White[2,*] ⓘ, Mar Aníbal-Martínez[3,*] ⓘ, Carmen Navarro-González[1,4] ⓘ, Cristina Gil-Sanz[2], Yaiza Domínguez-Canterla[1] ⓘ, Ana González-Manteiga[1], Verónica Del Buey Furió[1], Guillermina López-Bendito[3] ⓘ, Pietro Fazzari[1] ⓘ

**Schizophrenia is associated with altered cortical circuitry. Although the schizophrenia risk gene *NRG1* is known to affect the wiring of inhibitory interneurons, its role in excitatory neurons and axonal development is unclear. Here, we investigated the role of Nrg1 in the development of the corpus callosum, the major interhemispheric connection formed by cortical excitatory neurons. We found that deletion of Nrg1 impaired callosal axon development in vivo. Experiments in vitro and in vivo demonstrated that Nrg1 is cell-autonomously required for axonal outgrowth and that intracellular signaling of Nrg1 is sufficient to promote axonal development in cortical neurons and specifically in callosal axons. Furthermore, our data suggest that Nrg1 signaling regulates the expression of Growth Associated Protein 43, a key regulator of axonal growth. In conclusion, our study demonstrates that NRG1 is involved in the formation of interhemispheric callosal connections and provides a novel perspective on the relevance of NRG1 in excitatory neurons and in the etiology of schizophrenia.**

## Introduction

Schizophrenia (SZ) is a neurodevelopmental disorder that affects cognitive processes and social behavior (Harrison & Weinberger, 2005; Lewis & Sweet, 2009). Unlike other neuropathologies, the brains of SZ patients do not display obvious histological hallmarks. The most consistent endophenotypes in SZ include reduced neuropil, impaired functional connectivity between cortical areas (Mohr et al, 2000; Innocenti et al, 2003; Hoptman et al, 2012; Hennen et al, 2013; Arat et al, 2015; Fenlon & Richards, 2015), and specific changes in synaptic connections (Bjarnadottir et al, 2007; Li et al, 2007; Lisman et al, 2008; Lewis & Sweet, 2009). Therefore, SZ is

considered a pathology of abnormal wiring of cortical neurons. The structural–functional characterization of cortical connections is complex. However, the alterations observed in SZ can be conceptualized either as local alterations, for example, inhibitory feedback circuits (Bjarnadottir et al, 2007; Li et al, 2007; Lisman et al, 2008; Lewis & Sweet, 2009), or as long-range deficits involving connections between distant brain regions (Mohr et al, 2000; Innocenti et al, 2003; Fenlon & Richards, 2015). The corpus callosum (CC) is the largest bundle of cortico-cortical nerve fibers, and it connects the left and right cortical hemispheres. Converging evidence supports the hypothesis that the CC is hypoconnected in SZ (David, 1994; Fenlon & Richards, 2015). Specifically, morphological studies indicate a reduction of the CC and functional approaches, such as fMRI, show that interhemispheric coordination is impaired in SZ patients (Innocenti et al, 2003; Hoptman et al, 2012).

Notably, the development of CC is a complex process that begins during embryonic development, when callosal projecting neurons extend their axons medially, and continues into late postnatal stages of cortical development with the precise targeting of their contralateral counterparts (de León Reyes et al, 2020). Different guidance cues regulate the wiring of CC fibers, and the precise timing of CC development is critical for the proper formation of cortical circuits (Piper et al, 2009; Martín-Fernández et al, 2022).

Although the developmental etiology of SZ remains largely unresolved, it is well established that SZ has a strong genetic component. Numerous studies have identified the *neuregulin 1 (NRG1)* gene as a risk factor for the development of schizophrenia in various populations (Stefansson et al, 2002; Williams et al, 2003; Yang et al, 2003; Shyu et al, 2004; Tang et al, 2004; Harrison & Weinberger, 2005; Mei & Xiong, 2008). Interestingly, several studies in preclinical mouse models have shown that various genetic mutations that impair Nrg1/Erbb4 forward and intracellular signaling exhibit SZ-like symptoms, such as working memory deficits and hypersensitivity to psychostimulants (Stefansson et al, 2002;

[1]Lab of Cortical Circuits in Health and Disease, CIPF Centro de Investigación Príncipe, Valencia, Spain [2]Lab of Neural Development, BIOTECMED Institute, Universidad de Valencia, Valencia, Spain [3]Instituto de Neurociencias de Alicante, Universidad Miguel Hernández-Consejo Superior de Investigaciones Científicas (UMH-CSIC), Sant Joan d'Alacant, Spain [4]Department of Biotechnology, Universitat Politècnica de València, Valencia, Spain

Correspondence: pfazzari@cipf.es
Ana González-Manteiga's present address is Department of Radiation Oncology, Oria Laboratory, University of Cincinnati (US), Cincinnati, OH, USA
*Isabel Mateos-White and Mar Aníbal-Martínez contributed equally to this work

Coolen et al, 2005; Dejaegere et al, 2008; Mei & Xiong, 2008; Mei & Nave, 2014).

We and others have previously shown that the schizophrenia risk gene *Nrg1* is expressed in excitatory pyramidal neurons, whereas its specific receptor Erbb4 is mainly found in inhibitory interneurons. These studies demonstrated that Nrg1/Erbb4 signaling plays an important role in the cortex, and specifically in the wiring of inhibitory cortical neurons that express the Nrg1 receptor Erbb4 (Bjarnadottir et al, 2007; Li et al, 2007; Mei & Xiong, 2008; Chen et al, 2010a, 2010b; Fazzari et al, 2010, 2014; Pedrique & Fazzari, 2010; Rahman-Enyart et al, 2020; Navarro-Gonzalez et al, 2021). Erbb4 activation in inhibitory neurons is required for the proper wiring of local inhibitory circuits, as it promotes the growth of inhibitory axons in vitro and the formation of GABAergic synapses in vitro and in vivo (Fazzari et al, 2010; Rico & Marín, 2011; Navarro-Gonzalez et al, 2021). Moreover, acute stimulation with Nrg1 enhances the release of gamma-aminobutyric acid (GABA) from ErbB4-expressing interneurons (Woo et al, 2007; Mei & Xiong, 2008).

The vast majority of previous studies have focused on the role of Nrg1/Erbb4 signaling in interneurons. Conversely, the role of the reverse (intracellular) signaling of Nrg1 in cortical excitatory neurons is much less understood. In fact, the *Nrg1* gene encodes more than 30 isoforms and exhibits a complex bidirectional signaling. All isoforms contain the epithelial growth factor (EGF) domain, which binds and activates the ErbB4 receptor, stimulating the forward signaling (Bao et al, 2003; Mei & Xiong, 2008; Chen et al, 2010b; Fazzari et al, 2010; Pedrique & Fazzari, 2010; Rico & Marín, 2011; del Pino et al, 2013). In addition, most Nrg1 isoforms are transmembrane and contain the highly conserved intracellular domain (ICD), which mediates the Nrg1 intracellular signaling in pyramidal neurons. The physiological mechanisms leading to the activation of Nrg1 intracellular signaling are unclear, but in vitro studies have shown that it can be triggered by multiple stimuli, including binding to the ErbB4 receptor, neuronal depolarization, and hypoxia (Bao et al, 2003; Mei & Xiong, 2008; Navarro-González et al, 2019). Similar to APP and Notch, the regulated intramembrane proteolysis of Nrg1 culminates in the cleavage by gamma-secretase that releases the Nrg1-ICD in the cytosol, eventually followed by the translocation of the Nrg1-ICD to the nucleus (Bao et al, 2003; Chen et al, 2010b; Pedrique & Fazzari, 2010; Fazzari et al, 2014; Navarro-González et al, 2019).

In the current study, we investigated the role of Nrg1 in excitatory neurons and specifically in the development of callosal axons. Our in vivo experiments showed that Nrg1 signaling is required for the development of callosal connections in the mouse. Single-cell deletion of Nrg1 in primary neuronal cultures confirmed the role of Nrg1 in axonal development suggesting a cell-autonomous effect. Mechanistically, we found that the activation of Nrg1 intracellular signaling promotes axonal development in vitro and in vivo in callosal projecting neurons. In addition, we observed that the loss of Nrg1 reduced the expression of the Growth Associated Protein 43 (GAP43), a known enhancer of axonal growth (Chung et al, 2020). Interestingly, restoring GAP43 expression rescued axonal growth in Nrg1-deficient neurons in vitro, suggesting a potential downstream effector of Nrg1 signaling.

Altogether, our study indicates a role of Nrg1 intracellular signaling in the development of long-range cortico-cortical connections between brain hemispheres. These findings provide a novel perspective on the role of Nrg1 in the etiology of SZ. We propose that Nrg1 loss of function in excitatory neurons may contribute to the hypoconnectivity and neurodevelopmental alterations associated with SZ.

# Results

### Nrg1 regulates the development of callosal axons

The CC axons form the major interhemispheric connection. Their development is critical for cortical wiring, and interhemispheric connectivity is reduced in SZ (David, 1994; Fenlon & Richards, 2015). Here, we tested the hypothesis that the SZ risk gene *Nrg1* may control the development of interhemispheric cortical axons in the mouse. To investigate the development of callosal axons, we performed tracing experiments of callosal connections in newborn Nrg1 KO mice by labeling them with dye crystals (Figs 1A and S1A) (López-Bendito et al, 2006, 2007). At this stage, the expression of Nrg1 was strongly reduced in Nrg1 KO brains as compared to control littermates (Fig S1B). To trace cortico-cortical callosal axons, we placed small DiI and DiD crystals on the putative somatosensory cortices of control and Nrg1 KO brains. Consistent with previous studies (López-Bendito et al, 2007; Mizuno et al, 2007; Wang et al, 2007), dye tracing showed that in control mice, labeled somatosensory callosal axons approach the midline at P0 (Figs 1B and S1C and D). Notably, we found that in Nrg1 KO brains, the development of callosal axons is reduced as compared to control littermates. Specifically, we observed that in Nrg1 KO mice, the developing callosal axons were more distant from the midline in comparison with control axons (Fig 1B and C). We reasoned that this phenotype may be due to a reduced growth of callosal axons in Nrg1 KO cortices.

To rule out that this observation might be biased by a difference in the size or diffusion of the dye crystals, we measured the diffusion area of the dyes in the cortex (Fig S1C). We failed to show a relevant correlation between the cortical area labeled by the dye crystals and the distance of the axons from the midline (Fig S1E). Notably, at the rostro-caudal level that we traced, we found that the bundle of somatosensory callosal axons runs perpendicular to the anterior–posterior axis in both control and Nrg1 KO cortices (Fig S1D). This observation suggests that the rostro-caudal orientation of somatosensory callosal axons is not overtly impaired in Nrg1-deficient brains. Moreover, to exclude the possibility that the deficit in axon development was secondary to major impairments in earlier steps of callosal formation, we measured the thickness of the CC tract. We did not find obvious differences in the thickness of Nrg1 KO callosal structure compared with control littermates nor other major histological abnormalities, suggesting that CC development is delayed but not severely disrupted in the absence of Nrg1 (Fig S1F). Altogether, these results suggested that Nrg1 expression is required for proper development of callosal axons in vivo.

### Nrg1 is cell-autonomously required for the growth of cortical axons in vitro

We reasoned that the phenotype that we observed in Nrg1-deficient mice could be attributed to a direct role of Nrg1 in excitatory

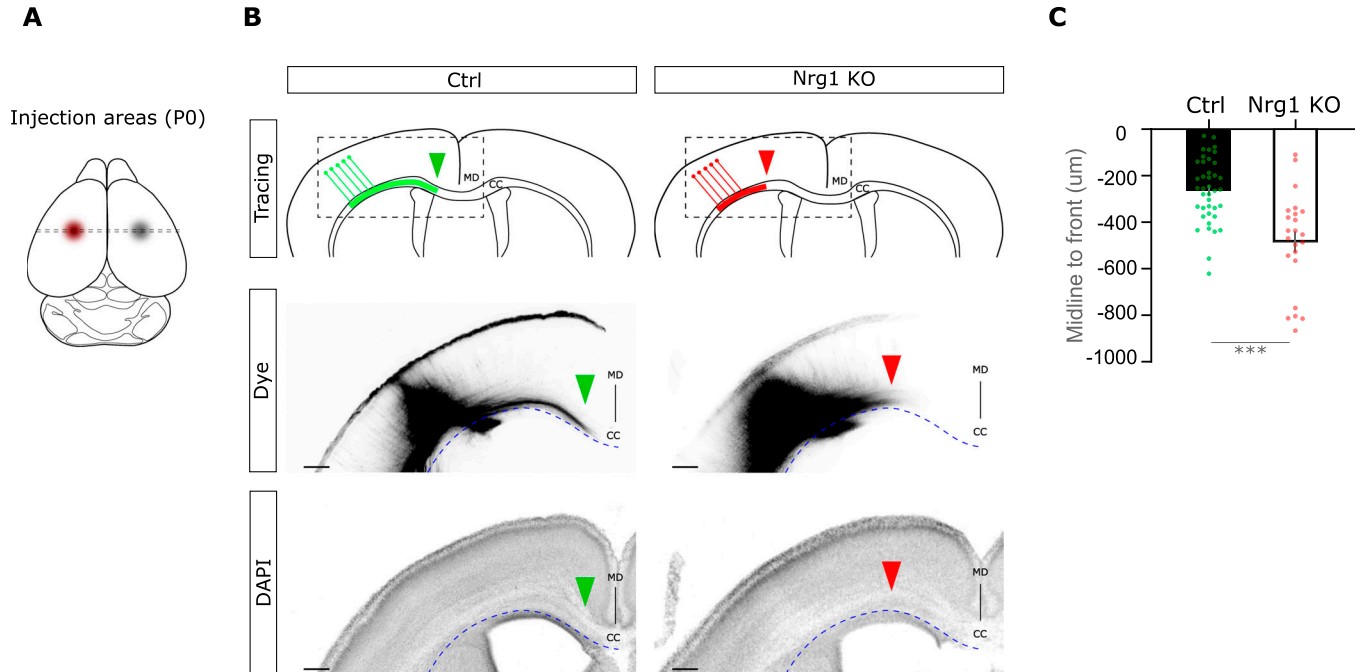

**Figure 1. Nrg1 is required for the development of cortico-cortical projecting axons in vivo.**
**(A)** Schema summarizing the experimental paradigm. In Nrg1 KO newborn mice, we carried out injections of DiI and DiD dye crystals (represented in red and gray, respectively) in both somatosensory cortices. The dashed lines indicate the rostro-caudal level. **(B)** Schematic representation and images of cortico-cortical projecting axons, both in Ctrl and in Nrg1 KO brains. Arrowheads indicate the axonal front within the corpus callosum. Boxed areas depict the magnified area in the images above. Dashed lines show the border of the CC. MD, midline; CC, corpus callosum. Scale bar, 200 $\mu$m. **(C)** Quantification of the position of the axonal front with respect to the midline (taken as the point zero). $n$ = 40 sections, out of eight brains, from three different litters for the Ctrl group, and $n$ = 23 sections, out of five brains, from three different litters for the Nrg1 KO condition. ***$P$ < 0.001, unpaired $t$ test. Average ± SEM.

neurons. However, cortical development is a complex process that requires the continuous interaction of multiple cell types in the brain (de León Reyes et al, 2020). Because Cre expression under the Nestin promoter drives Nrg1 deletion in all brain cells (Tronche et al, 1999), alterations in non-neuronal cell types might in principle cause the deficits in the development of callosal connections that we observed. Therefore, to directly determine the role of Nrg1 in axonal development, we took advantage of a more reductionist in vitro model.

We established primary cultures of cortical neurons from Nrg1$^{flox/flox}$ mice (Fig S2A). To obtain a spare labeling and single-cell resolution, we performed co-culture of naïve Nrg1$^{flox/flox}$ neurons with Nrg1$^{flox/flox}$ neurons expressing Cre to obtain Nrg1-deficient neurons. This experimental paradigm allowed us to evaluate the effect of Nrg1 deletion in cortical neurons developing together with non-mutant cells. We found that axonal length was significantly reduced in Nrg1 KO neurons as compared to control cells (Figs 2A and B and S2D). In contrast, the branch density (number of axonal branches per length) was not affected by the loss of Nrg1 (Fig 2C). Besides, Sholl analysis showed a reduction in dendritic extension (Fig S2B and C).

Taken together, these observations suggest that Nrg1 is required for the development of cortical axons in a cell-autonomous manner. This interpretation is consistent with the hypothesis that the deficit in callosal axonal growth may be caused by the deletion of Nrg1 in callosal projecting neurons in Nrg1 KO mice (Fig 1).

## Nrg1 intracellular signaling promotes axonal growth

Because Nrg1 loss of function impaired axonal growth (Figs 1 and 2), we next asked whether Nrg1 expression was sufficient to promote axonal development. To address this point, we performed gain-of-function experiments in a single-cell experimental paradigm similar to the one described above; namely, we cultured naïve primary cortical neurons with neurons transfected to express type III Nrg1, hereafter Nrg1-FL (Fig 3A–D). Nrg1-FL is one of the major isoforms of Nrg1 and, like most Nrg1 isoforms, contains a transmembrane domain and a long ICD (Fig 3B). Notably, the expression of Nrg1-FL in cortical neurons increased axonal elongation as compared to control neurons (Fig 3A and C). Nrg1-FL undergoes a stepwise processing, which ends with the cleavage of the Nrg1 transmembrane domain by gamma-secretase. As a result, the ICD of Nrg1 is released in the cytosol and it translocates to the nucleus (Bao et al, 2003; Chen et al, 2010b; Fazzari et al, 2014; Navarro-González et al, 2019). To determine the role of Nrg1 intracellular signaling in axonal elongation, we expressed Nrg1-ICD in cortical neurons. We and others (Bao et al, 2003; Fazzari et al, 2014; Navarro-González et al, 2019) previously showed that the expression of Nrg1-ICD effectively mimics the activation of Nrg1 intracellular signaling. Here, we found that the expression of Nrg1-ICD was sufficient to promote axonal growth in developing cortical neurons (Figs 3A and C and S3C). The expression of either Nrg1-FL or Nrg1-ICD did not

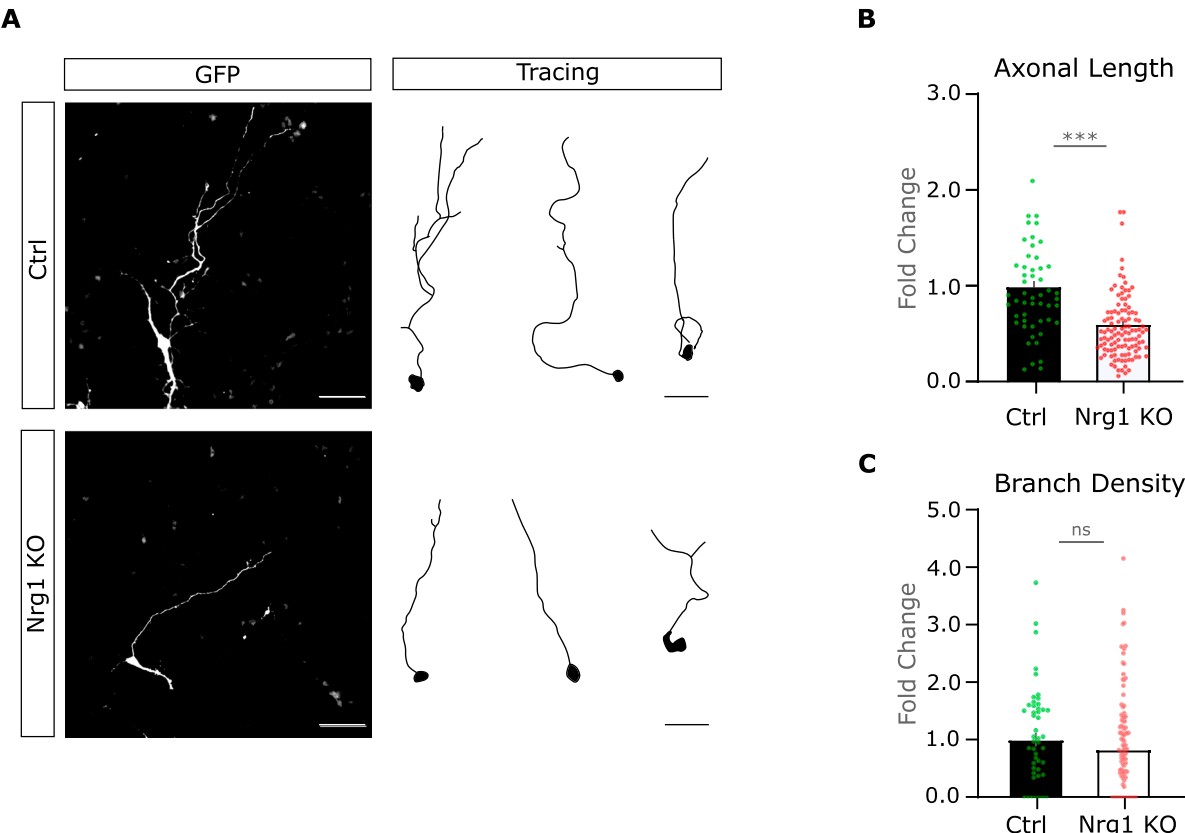

**Figure 2. Nrg1 is necessary for axonal growth in vitro.**
**(A)** Representative pictures and drawings of control and Nrg1 KO cultured neurons. The neurons were electroporated to express GFP-ires-Cre to perform single-cell deletion to obtain Nrg1 KO neurons and co-cultured with control neurons. Scale bar, 50 μm. **(B)** Graph illustrates the quantification of the axonal length expressed in fold change as compared to the control. Ctrl: *n* = 52; Nrg1 KO: *n* = 116, out of three independent litters. ***$P < 0.001$, unpaired *t* test. Average ± SEM. **(C)** Graph showing the quantification of axonal branch density. Ctrl: *n* = 52; Nrg1 KO: *n* = 116, both out of three independent litters. Unpaired *t* test, $P = 0.26$. Average ± SEM.

significantly alter axonal branching density (Fig 3D) or dendrite growth (Fig S3A and B). These findings suggest a molecular mechanism by which Nrg1 expression cell-autonomously promotes axonal development by activating Nrg1 intracellular signaling.

### GAP43 is a putative effector of Nrg1 signaling in axonal growth

Because Nrg1 loss reduced axonal growth, we hypothesized that Nrg1 deletion might affect the expression of key regulators of axonal development. To further explore the mechanisms behind Nrg1 signaling, we examined the effect of Nrg1 loss on key pathways involved in axonal growth, including AKT, JNK, ERK, and GAP43 (O'Donnell et al, 2009; Chung et al, 2020). Consistent with our hypothesis, Nrg1 ablation in primary cortical neurons from Nrg1 KO mice resulted in a significant decrease in GAP43 expression levels as compared to control littermates (Fig 4A and B). In contrast, our results suggest that Nrg1 deletion does not overtly affect the expression and activation of the AKT, JNK, and ERK signaling pathways in our experimental model (Fig S4A and B).

To functionally assess the role of GAP43 in Nrg1 signaling, we examined its ability to rescue the growth defects observed in Nrg1-deficient neurons (Figs 4C and D and S4C–E). Notably, we found that

restoring GAP43 expression in these neurons cell-autonomously rescued axonal growth in Nrg1-deficient neurons (Figs 4C and D and S4E). Taken together, these findings suggest that GAP43 may be a relevant downstream effector of Nrg1 signaling in promoting axonal growth in cortical neurons.

### Nrg1 signaling promotes the development of callosal axons in vivo

We have shown that Nrg1 is required for the development of interhemispheric projections (Fig 1), and our in vitro experiments suggested that Nrg1 signaling cell-autonomously promotes axonal elongation in vitro (Fig 3). Therefore, we next asked whether the activation of Nrg1 signaling was sufficient to promote the outgrowth of the callosal axons also in vivo. To determine the role of Nrg1 signaling in vivo, we performed gain-of-function experiments using in utero electroporation (*IUE*). This experimental approach is particularly effective for studying the development of cortico-cortical callosal axons because it allows targeting specifically the upper cortical layers that contain most of the contralateral projecting cortical neurons (de León Reyes et al, 2020; Mateos-White et al, 2020). Therefore, we performed IUE at E15.5 to obtain the expression of Nrg1 in layer 2/3 callosal

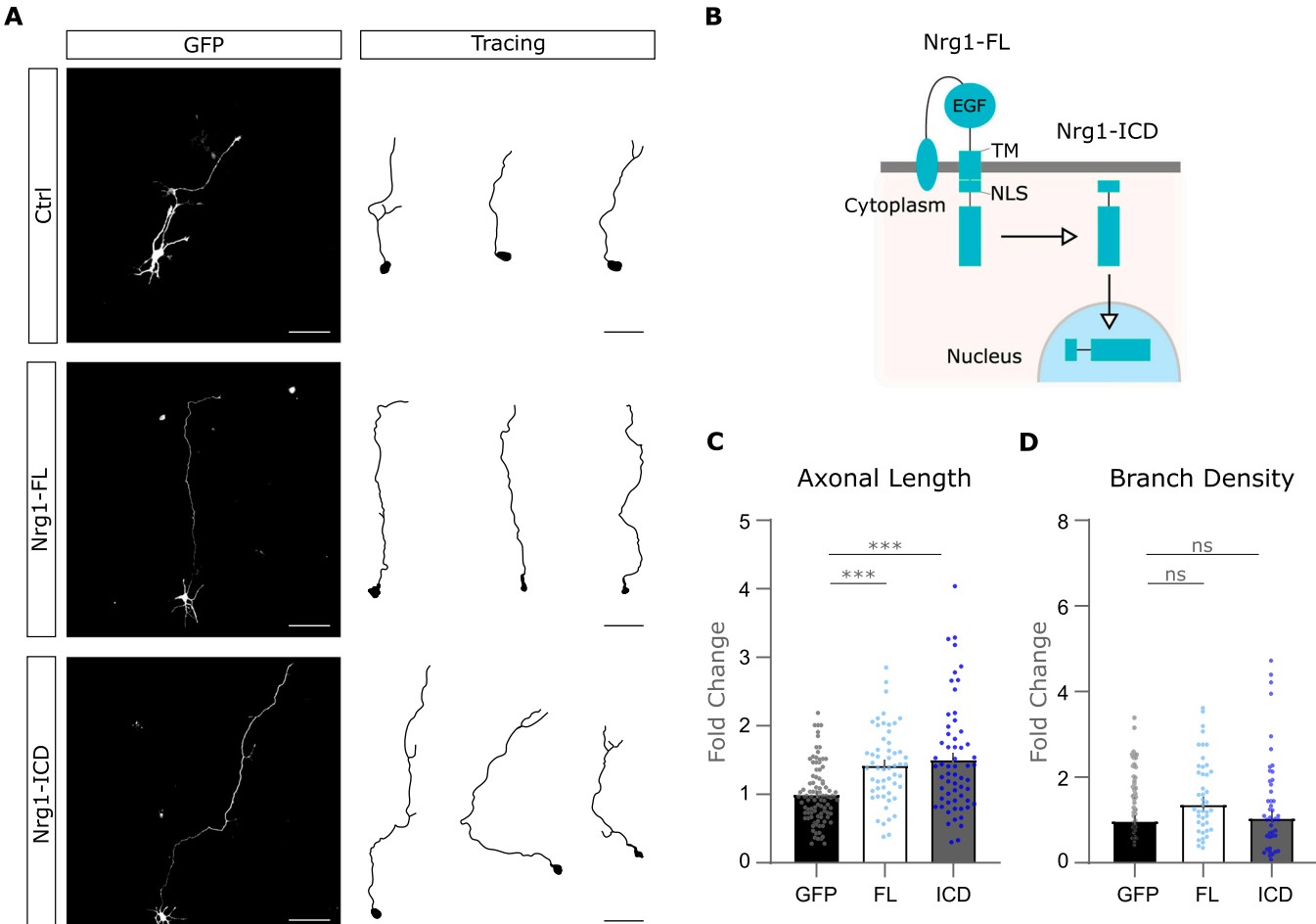

**Figure 3. Nrg1 intracellular signaling is sufficient to promote axonal growth in vitro.**
**(A)** Representative images and drawings of control and Nrg1-overexpressing cultured neurons. The neurons were transfected to express either Nrg1-FL-GFP or Nrg1-ICD-GFP and co-cultured with control neurons. The cells were fixed at DIV4. Scale bar, 50 μm. **(B)** Schematic representation of the structure of Nrg1-FL: on the left in its full conformation (also named Nrg1-FL), which includes an epithelial growth factor domain, a transmembrane domain (TM), a cysteine-rich domain, and an NLS; on the right, the intracellular Nrg1 (Nrg1-ICD). **(C)** Quantification of the axonal length of Nrg1-expressing neurons shown in fold change as compared to the control. Ctrl: $n$ = 96; Nrg1-FL–overexpressing neurons: $n$ = 55; Nrg1-ICD–overexpressing neurons: $n$ = 60, out of three independent neuronal cultures. One-way ANOVA and post hoc with Turkey's test, ***$P$ < 0.001. Average ± SEM. **(D)** Graph showing the quantification of axonal branch density control and Nrg1-FL– and Nrg1-ICD–expressing neurons. Ctrl: n = 78, Nrg1-FL–overexpressing neurons: n = 55; Nrg1-ICD–overexpressing neurons: n = 60, all of them out of three independent neuronal cultures. One-way ANOVA and post hoc with Tukey's test. Ctrl versus Nrg1-FL, $P$ = 0.27; Ctrl versus Nrg1-ICD, $P$ = 0.94. ns, not significant. Average ± SEM.

neurons (Fig S5A) (Mateos-White et al, 2020). Specifically, we electroporated cortical neurons to express either Nrg1-FL or GFP as a control. Because the exact timing of axonal development may vary from litter to litter, we expressed Nrg1-FL or GFP in littermates to provide an internal control (Fig 5A–C). We performed the analysis of electroporation at P2 because in our experimental settings, the electroporated callosal axons approach the midline at this stage. We measured the distance between the electroporation zone and the more advanced axons in the corpus callosum (Fig S5B). Notably, we found that the expression of Nrg1-FL increased axonal elongation in callosal projecting neurons as compared to control littermates (Figs 5A–C and S5C). To investigate the signaling mechanism involved in this process, we also performed gain-of-function experiments expressing Nrg1-ICD to selectively determine the role of Nrg1 intracellular signaling (Figs 5D–F and S5D). Consistent with the

in vitro experiments (Fig 3), we found that the activation of Nrg1 intracellular signaling was sufficient to promote the growth of callosal axons. Altogether, these results suggest Nrg1 promotes the formation of interhemispheric callosal connections by activating Nrg1 intracellular signaling.

Finally, we investigated whether Nrg1 loss in the adult stage could affect the callosal projections. To this aim, we analyzed the pattern of callosal projections in UBC-CreER2; Nrg1[flox/flox] after acute ablation of Nrg1 by tamoxifen treatment (Fig S6A–D). Our analysis did not reveal significant differences in the profile of callosal connections between Nrg1-ablated and control mice. These results suggest that Nrg1 may play a redundant role in the maintenance of established callosal projections. Future studies will be needed to determine whether Nrg1 deletion in adulthood affects other aspects of callosal function, such as synaptic transmission or plasticity.

**Figure 4. GAP43 expression is reduced by Nrg1 loss and rescues axonal growth in Nrg1-deficient neurons.**
**(A)** Western blot (WB) analysis of GAP43 protein levels in control (Nrg1$^{flox/flox}$) and Nrg1-deficient (Nes-Cre; Nrg1$^{flox/flox}$) neurons at DIV4. Tubulin levels are shown as a loading control. **(A, B)** Quantification of GAP43 protein levels from (A). GAP43 protein levels are normalized to tubulin. Data are represented as the mean ± SEM. Statistical significance was determined using an unpaired $t$ test with Welch's correction. Ctrl: n = 6; Nrg1 KO: n = 5, mice, from two litters (*$P < 0.05$). **(C)** Representative images and

# Discussion

Our study identifies a novel role of Nrg1 intracellular signaling in excitatory neurons. Specifically, we showed that Nrg1 is necessary and sufficient to promote the axonal development of callosal projections in vivo. Our in vitro experiments in primary cortical neurons showed that single-cell deletion of Nrg1 impairs the development of the axons. Conversely, the single-cell expression of Nrg1-FL or Nrg1-ICD, to selectively activate Nrg1 intracellular signaling, was sufficient to promote axonal growth. Tracing experiments in Nrg1-deficient mice showed that the development of callosal axons is altered, indicating that Nrg1 signaling is critical for promoting the formation of callosal connections. Moreover, gain-of-function experiments showed that the expression of Nrg1 full length or its ICD enhanced the development of callosal axons in vivo. Altogether, these results provide the first evidence that Nrg1 is important for the development of cortical axons and callosal projections in vitro and in vivo.

The delay in the growth of cortical callosal connections observed in Nrg1-deficient mice could have significant functional and behavioral consequences. Interhemispheric integration of brain function depends largely on the wiring of the callosal axons, and the development of precise contralateral connections is critical for most aspects of cortical function, including basic brain activities and higher cortical functions. For instance, callosal wiring is crucial for coordinating motor actions, creating three-dimensional representations of visual and auditory stimuli, processing emotions, and generating verbal responses (Fenlon & Richards, 2015; de León Reyes et al, 2020). Interestingly, in humans, higher order cortical regions such as Broca's and Wernicke's areas are lateralized asymmetrically. These regions control writing and speech, and their functions are impaired by surgical corpus callosotomy (Mohr et al, 2000; Riès et al, 2016).

The formation of CC projections involves a tightly regulated sequence of stepwise events (Fenlon & Richards, 2015). This process begins in the embryo with the specification of contralateral projecting neurons and the axonal extension toward the midline, continuing postnatally with the innervation of the contralateral side. In mice, CC axons reach their contralateral target and arborize around P7–P10 (Mizuno et al, 2007; Wang et al, 2007), a critical period for cortical wiring that involves a peak in synaptogenesis and the concomitant activity-dependent apoptosis of cortical inhibitory interneurons (Wong et al, 2018; Favuzzi et al, 2019). Finally, redundant CC connections are pruned after weaning until P30 (De León Reyes et al, 2019). Because of this complex time-locked integration of interhemispheric and local intracortical wiring, the developmental delay of callosal axons observed in Nrg1-deficient mice may lead to long-lasting deficits in interhemispheric wiring, potentially affecting cognitive and behavioral functions. Future

studies, such as electrophysiological recordings to determine the interhemispheric correlation of neuronal activity, will be required to more specifically address the functional consequences of Nrg1 loss.

Given its major functional relevance and complex development, it is not surprising that CC deficits were associated with neurodevelopmental disorders (David, 1994; Innocenti et al, 2003; Hoptman et al, 2012; Fenlon & Richards, 2015). The cellular and molecular underpinnings of the alterations of CC development in SZ remain largely undetermined. The most straightforward explanation is that a reduced development of CC would directly impair interhemispheric communication and information processing. However, we cannot exclude more complex sequelae of an altered CC wiring. For instance, because CC extension and wiring of the contralateral targets occur at a crucial stage for the remodeling of cortical circuitries (around P7–P10 in mice) (Mizuno et al, 2007; Wang et al, 2007), it is possible that the CC plays a crucial role in this process. Several studies have shown that Nrg1-deficient mice exhibit SZ-related behavioral changes. However, given the multiple roles that Nrg1 plays in brain development, it will be very difficult to determine the specific contribution of the deficits we observed in callosal development to the behavioral alterations. Nevertheless, we speculate that the delay in callosal development may contribute, at least in part, to the SZ-like symptoms in Nrg1 mutant mice.

Because *Nrg1* was identified as a major SZ risk gene (Stefansson et al, 2002; Hennen et al, 2013), most studies attempted to understand the role of Nrg1 in brain wiring and in cortical inhibition. Altogether, these studies showed that Nrg1 plays a pivotal role in various steps of cortical development and in the wiring of the inhibitory circuits (Chen et al, 2010a, 2010b; Fazzari et al, 2010; Pedrique & Fazzari, 2010; Rahman-Enyart et al, 2020; Navarro-Gonzalez et al, 2021).

With regard to cortical interneurons, it was shown that Nrg1 controls the migration of interneuron precursors from the ganglionic eminence to the cortex (Flames et al, 2004). Postnatally, Nrg1 is important for the activity of cortical interneurons and inhibitory homeostasis (Mei & Xiong, 2008; Mei & Nave, 2014). Indeed, Nrg1 was one of the first synaptogenic cues found to regulate cortical inhibitory circuits via the activation of Erbb4. Notably, most studies investigated the role of the forward (or canonical) Nrg1/Erbb4 signaling in interneurons. Specifically, we and others found that Nrg1 promotes, in ErbB4-expressing inhibitory cells, the formation of excitatory synapses in the dendrites of inhibitory contacts in the axonal button (Mei & Xiong, 2008; Chen et al, 2010a; Navarro-Gonzalez et al, 2021). Besides, Nrg1 regulates the development of dendritic spines in pyramidal neurons, probably via its intracellular signaling (Barros et al, 2009; Fazzari et al, 2014). Interestingly, although most studies in preclinical models have focused on the loss of Nrg1/ErbB4 signaling, others have shown that the exogenous

corresponding tracings of immunofluorescence staining in control, Nrg1-deficient (Nrg1 KO), and Nrg1 KO neurons expressing GAP43 (Nrg1 KO+GAP43) at DIV4. Scale bar, 50 *μ*m. **(D)** Quantification of axonal length in control, Nrg1 KO, and Nrg1 KO+GAP43 neurons. Data are represented as the mean ± SEM. Statistical significance was determined using a one-way ANOVA test with Tukey's post hoc test. Ctrl: n = 104; Nrg1 KO: n = 105; GAP43: n = 78, neurons from two independent experiments (**$P < 0.01$, ***$P < 0.001$). **(E)** Quantification of branch density in control, Nrg1 KO, and Nrg1 KO+GAP43 neurons. Data are represented as the mean ± SEM. Statistical significance was determined using a one-way ANOVA test with Tukey's post hoc test. Ctrl: n = 93; Nrg1 KO: n = 98; GAP43: n = 73, neurons from two independent experiments. ns, not significant.

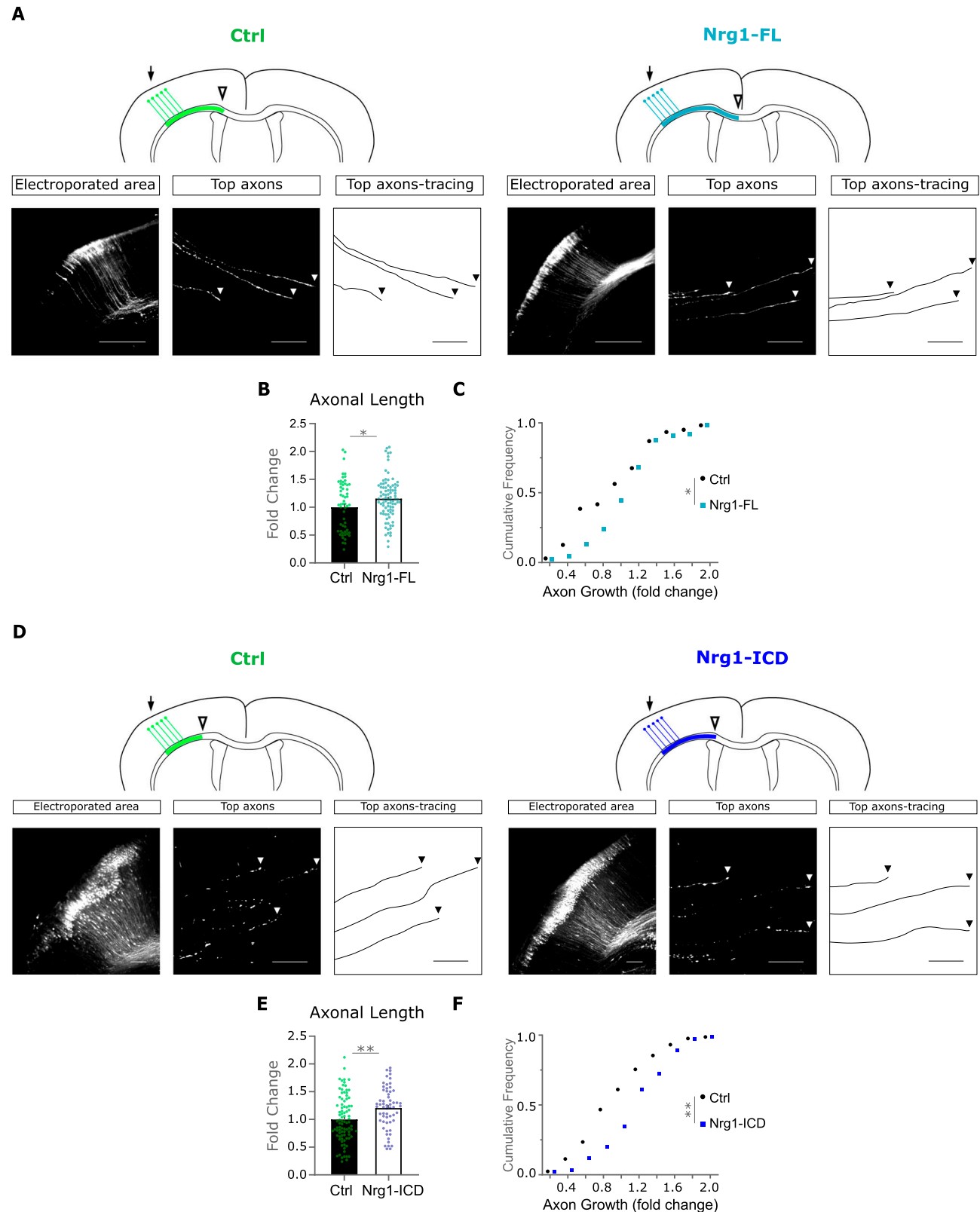

**Figure 5. Nrg1 signaling enhances the elongation of callosal axons in vivo.**
**(A)** Schematic representation of developing callosal axons at P2 in control littermates and in neurons expressing Nrg1-FL. Arrows indicate the electroporated area, and the empty arrowheads show the location of the more advanced axonal tips (top axons). The images below show the electroporated area, the top axons, and their tracing. The arrowheads indicate the tip of the axons. Scale bar, 50 μm. **(B, C)** Quantification of axonal extension in Nrg1-FL–expressing neurons relative to control littermates. **(B, C)**

expression of Nrg1 can also be detrimental to cortical wiring and lead to SZ-like symptoms (Hahn et al, 2006; Yin et al, 2013; Agarwal et al, 2014; Olaya et al, 2018). These results suggest that an optimal level of Nrg1 is required to maintain homeostasis of excitatory/ inhibitory circuits in the cortex (Agarwal et al, 2014).

The role of Nrg1 in axonal development in pyramidal neurons is poorly understood. Nonetheless, a few studies suggested that Nrg1 loss of function may impair dendritic development (Zhang et al, 2016, 2017). In particular, constitutive deletion of the type III isoform of Nrg1 showed reduced dendritic arborization in mouse embryos (Chen et al, 2010b). Moreover, primary cortical neurons from this type III Nrg1 mutant showed impaired dendrite development at DIV3, an early stage of maturation. Nrg1 intracellular signaling could partially rescue the developmental deficits in dendrites (Chen et al, 2010b).

Our work showed that Nrg1 intracellular signaling is required for axonal development in vitro. Notably, in our experimental settings, the Cre-dependent deletion of Nrg1 is acute. Therefore, we can reasonably rule out the caveat of possible early developmental deficits unrelated to the specific process of axonal elongation. Moreover, we showed, to our knowledge for the first time, that Nrg1 signaling is necessary and sufficient for the growth of cortical axons in vivo, specifically in cortico-cortical callosal connections.

Mechanistically, our experiments indicated that the activation of Nrg1 intracellular signaling is involved in axonal growth both in vitro in primary cultures and in vivo in callosal projecting neurons. This growth-promoting activity might be cell-autonomous, as activating Nrg1 intracellular signaling in single cells through the expression of Nrg1-ICD is enough to stimulate axonal growth in vitro and in vivo. The direct downstream effectors of Nrg1 intracellular signaling are still unknown. Our data showed that Nrg1 deletion in neuronal cultures led to a significant decrease in the expression of GAP43, a well-known player in axonal growth and regeneration. Interestingly, GAP43 expression could cell-autonomously rescue the decrease in axonal development in Nrg1 KO primary cortical neurons in vitro. These findings suggest that GAP43 may be a relevant mediator of Nrg1 signaling in axonal development. Future studies will be required to further investigate the growth-associated pathways downstream of Nrg1 signaling.

In conclusion, our work showed a novel role of Nrg1 in the development of callosal axons, the major contralateral connection in the brain. Given the importance of callosal connections for brain function in physiological conditions and in SZ, we speculate that this alteration, together with the previously reported synaptic deficits, may contribute to the behavioral phenotype of Nrg1-deficient mice. Altogether, our study provides a novel perspective on the role of Nrg1 and its intracellular signaling in SZ, highlighting the need for further research into the functional and behavioral impacts of disrupted Nrg1 signaling on cortical connectivity.

# Materials and Methods

## Animals

In this study, we crossed mice expressing Cre under Nestin promoter (B6.Cg-Tg[Nes-cre]1Kln/J, hereafter Nes-Cre; kindly provided by Professor Rüdiger Klein, Max Plank Institute for Biological intelligence) with conditional mutant mice for Nrg1 (Nrg1$^{tm3Cbm}$; aka Nrg1$^{flox/flox}$; MGI:2447761) that carry a "floxed" Nrg1 allele. In Nes-Cre; Nrg1$^{flox/flox}$ mice (Nrg1 KO), the embryonic expression of Cre in the central nervous system results in the developmental deletion of the exons 7–9, which abrogates the expression and the Nrg1 signaling starting from embryonic day 11 (Tronche et al, 1999). Nes-Cre; Nrg1$^{flox/flox}$ mice were viable, were fertile, and did not show major signs of alterations in our housing conditions. For neuronal cultures, Nes-Cre; Nrg1$^{flox/flox}$ mice were crossed with Nrg1$^{flox/flox}$ mice to obtain Nrg1 KO and control Nrg1$^{flox/flox}$ mice from the same litter. CD-1 background mice were used for the Nrg1 overexpression studies in vitro, and C57BL/6J females and pups for the gain or loss of function in vivo, respectively. The Nrg1$^{flox/flox}$ conditional mouse model was used to study the effect of Nrg1, by producing a single-cell loss-of-function model in vitro with the expression of a Cre-containing plasmid in cultured neurons. The Nes-Cre transgenic mice were employed for the loss of function in vivo, as well as the Nrg1 molecular mechanism studies except for Fig S6. For Fig S6, we crossed Nrg1$^{flox/flox}$ with UBC-CreER2 mice (B6.Cg-Ndor1Tg[UBC-cre/ERT2]1Ejb/2J mice; Jax strain #008085; kindly provided by Professor Eric Brown). We performed the experiments in 12-mo-old female mice using littermates as a control. The animals were kept and cared for in the animal facility of the CIPF Centro de Investigación Príncipe Felipe (Valencia, Spain). The experiments were supervised by the Bioethics Committee of the Institute and performed in compliance with bioethical regulations of the European Union and Spain. Animals were group-housed with food and water ad libitum in standard housing conditions.

## Nrg1 constructs

The original constructs for the expression of GFP-tagged Nrg1-ICD and Nrg1-FL, from *Mus musculus*, were previously generated and fully described (Fazzari et al, 2014). Briefly, Nrg1-FL expresses the CRD-Nrg1 isoform of Nrg1, aka type III Nrg1. Nrg1-ICD expresses the entire ICD of CRD-Nrg1 including the nuclear localization signal. Nrg1-ICD therefore mimics the ICD of Nrg1 resulting from the cleavage by gamma-secretase. Nrg1 is expressed under the CMV promoter for in vitro experiments and the CAG promoter for IUE (Fazzari et al, 2014).

---

Bar graph shows the mean ± SEM (B), whereas the plot shows the cumulative frequency (C). Ctrl: $n = 60$ axons, out of six sections from four brains; Nrg1-FL: $n = 90$ axons, out of nine sections from four brains. **(B)** Unpaired $t$ test, *$P < 0.05$. **(C)** Kolmogórov–Smirnov test, *$P < 0.05$. **(D)** Drawing shows the callosal axons at P2 in control littermates and in Nrg1-ICD–expressing neurons. Arrows indicate the electroporated area, and the empty arrowheads show the location of the more advanced axonal tips (top axons). The pictures show representative images of the electroporated area, the top axons, and their tracing. The arrowheads indicate the tip of the axons. Scale bar, 50 µm. **(E, F)** Bar graph shows the mean ± SEM (E), whereas the plot shows the cumulative frequency (F). Ctrl: $n = 90$ axons, out of nine sections from three brains; Nrg1-ICD: $n = 60$ axons, out of six sections from three brains. Both conditions are from one litter, counting with internal controls within littermates. **(E)** Unpaired $t$ test, **$P < 0.01$. F, Kolmogórov–Smirnov test, **$P < 0.01$.

 **Life Science Alliance**

## Characterization of Nrg1 deletion by qRT-PCR

We did neuronal cultures from embryos at E15 as described below. Briefly, the difference is that embryos are genotyped and cells are plated individually, obtaining two wells in six-well plates, with 400,000 cells in each one. Neurons were cultured in normoxic conditions and analyzed at the fourth day in vitro (DIV). Neuronal cultures were homogenized with TRIzol (15596018; Thermo Fisher Scientific), and RNA was extracted with Direct-zol RNA Miniprep Kits (R2052; Zymo) following the manufacturer's instructions. Newborn mice (P0) were transcardially perfused with PBS as previously described (Fazzari et al, 2014). One hemicortex was used for mRNA extraction as follows. RNA was extracted from the cortex according to the manufacturer's instructions, using TRIzol (15596018; Thermo Fisher Scientific) and a FastPrep 24 5G homogenizer (6005500; MP Biomedicals). RNA was quantified by absorbance at 260 nm using a NanoDrop ND-100 (Thermo Fisher Scientific). The cDNA was prepared using the High Capacity cDNA Reverse Transcription kit (4368814; Thermo Fisher Scientific), and the quantitative PCR (qRT-PCR) was performed with TB Green Premix Ex Taq (Tli RNase H Plus) 2x (RR420; Takara Bio) for Nrg1-EGF, CRD-Nrg1, Nrg1-TM, Rbfox3, and Gapdh primer pairs, in a LightCycler 480 (Roche). The Ct was calculated in LightCycler 480 software. All values were normalized to the housekeeping gene Gapdh and to the mean value of the control samples.

The primers used for qRT-PCR amplification were as previously described (Navarro-González et al, 2019; Navarro-Gonzalez et al, 2021). The primer pairs used in this study were designed and validated by PrimerBank (https://pga.mgh.harvard.edu/primerbank/) as previously described (Navarro-González et al, 2019; Navarro-Gonzalez et al, 2021), except for Nrg1-EGF and Nrg1-CRD primer pairs, which were previously published in Makinodan et al (2012) (PMID: 22984073).

## Primary neuronal culture and neuron transfection by electroporation

Primary cultures of cortical neurons were prepared from embryonic day 15–16 (E15–16) of CD-1 or C57BL/6 (Nrg1$^{flox/flox}$ and Nes-Cre) mice, as previously described (Navarro-González et al, 2019; Rodríguez-Prieto et al, 2021). Briefly, embryonic brains were dissected and placed into ice-cold Hank's solution (14175-095; Thermo Fisher Scientific) with 7 mM Hepes and 0.45% glucose (15630-080; Thermo Fisher Scientific). The tissue was trypsinized (25300054; Thermo Fisher Scientific) at 37°C for 15 min. Cortices were washed with Hank's solution and dissociated by mechanical disaggregation in 5 ml of plating medium (MEM supplemented with 10% horse serum and 20% glucose). Cells were counted in a Neubauer chamber and plated into precoated dishes with poly-D-lysine (P2636; Sigma-Aldrich), for a final concentration of 110,000 cells per well in 12-well plates. The rest of the cells were transfected to express the gene of interest when needed. Transfection was performed with the NEPA21 electroporation system as previously described (Rodríguez-Prieto et al, 2021). Briefly, after disaggregation, the adequate volume of the cell suspension was transferred into a new tube, centrifuged, and resuspended in electroporation medium (Opti-MEM 31985-062; Invitrogen). The final volume corresponding to 1 million of cells was mixed in a cuvette with the

desired amount of DNA per electroporation condition: pCMV-GFP, pLV-CMV-LoxP-DsRed-LoxP-eGFP (65726; Addgene; kindly provided by Dr. Jacco van Rheenen), pLenti-Lifeact-tdTomato, pCMV-GFP-ires-Cre (Fazzari et al, 2010), pCAG-Nrg1-FL-nGFP, pCAG-Nrg1-ICD-GFP (Bao et al, 2003; Fazzari et al, 2014). The cuvettes were inserted in the electroporator under the pre-established conditions: poring pulse of 2 ms, 175 V, 50 interval, 10 decay rate, positive polarity, two times; transfer pulse of 50 ms, 20 V, 50 interval, 40 decay rate, positive and negative polarity, five times (Rodríguez-Prieto et al, 2021). Transfected cells were mixed with 0.5 ml of plating medium and co-cultured with the previously plated non-electroporated neurons. The cells were placed into a humidified incubator containing 95% air and 5% CO2. Two hours after plating, the plating medium was replaced with equilibrated Neurobasal media supplemented with B27 (17504-044; Life Technologies) and GlutaMAX (35050038; Thermo Fisher Scientific). For all experiments in vitro, all the neurons were electroporated to express GFP to visualize the neuronal structure with the immunolabeling anti-GFP. For both Nrg1 loss-of-function and gain-of-function experiments, control neurons were co-cultured together with Nrg1 KO– and Nrg1-FL/Nrg1-ICD–overexpressing neurons in the same well to have an optimal internal control. Control neurons were co-transfected with pCMV-GFP to visualize the morphology and with either pLV-CMV-LoxP-DsRed-LoxP-eGFP or pLenti-Lifeact-tdTomato to identify the control neurons in the red channel. In our experimental conditions, the expression of either DsRed or Lifeact-tdTomato did not affect axonal growth and was simply used to distinguish control cells from Nrg1 KO– and Nrg1-FL/Nrg1-ICD–overexpressing neurons. Nrg1 KO neurons were obtained by transfecting Nrg1 conditional neurons with pCMV-GFP-ires-Cre to visualize the morphology in the green channel and to express Cre to abrogate Nrg1 expression. For gain-of-function experiments, wild-type neurons were co-transfected to express pCMV-GFP to visualize neuronal morphology together with either Nrg1-FL or Nrg1-ICD. Neurons were fixed at DIV4 with 4% PFA in PBS. Immunofluorescence was carried out as follows in the next paragraph.

## Protein extraction from neuronal cultures and Western blot and quantification

For Nrg1 signaling studies, protein extraction and Western blotting were carried out. Primary cultures of cortical neurons were prepared from E15–16 of Nrg1 Nes-Cre mice, as described above. Briefly, the difference is that embryos are genotyped and they are plated individually, obtaining three wells in six-well plates with 400,000 cells for each one. Neurons were cultured in normoxic conditions and analyzed at DIV5.

The plated neurons were homogenized with RIPA 1x buffer (150 mM NaCl, 1% Nonidet P-40, 50 mM Tris, 0.5% sodium deoxycholate, 0.1% SDS) containing protease inhibitors (1 mM PMSF, 0.1 mM leupeptin, 2 mM Na2VO4, 100 mM NaF, 20 mM Na4P2O7), following the General Protocol for Western blotting (Bio-Rad) as previously described (Navarro-Gonzalez et al, 2021). An equal amount of denatured samples were run on 12% acrylamide SDS–PAGE. Proteins from gels were transferred onto PVDF membranes (03010040001; Roche), and they were incubated with the primary antibodies GAP43 (1:5,000, Cat#: GAP43; Aves Lab); p-SAPK/JNK

(T183/Y185) (1:1,000, 9251S; Cell Signaling); SAPK/JNK (1:1,000, 9252S; Cell Signaling); p44/42 MAPK (Erk1/2) (1:1,000, 9102S; Cell Signaling); pERK (1:1,000, sc-7383; Santa Cruz); p-Akt (S473) (1:1,000, 4060S; Cell Signaling); Akt, (9272S; Cell Signaling); and tubulin-HRP (1:50,000, AC030; ABclonal), diluted in 1% BSA/TBST buffer, at 4°C overnight. The primary antibodies were blotted with HRP-conjugated secondary antibodies, and both signals were developed using an ECL chemiluminescence detection kit (NEL105; PerkinElmer Life Sciences). Signals were detected and imaged with the luminescent image analyzer Uvitec Q9 Alliance and quantified with Image-QuantTL software. Protein levels were normalized to tubulin as a loading control.

### Immunofluorescence

Immunocytochemistry was performed according to a standard protocol previously described (Navarro-González et al, 2019). Briefly, neurons were fixed with 4% PFA for 10 min, permeabilized for 10 min with PBS/0.1% Triton, and then blocked with 2% PBS/BSA. Primary and secondary antibodies were diluted in 2% PBS/BSA. The primary antibody incubation was made during 24 h at 4°C and the secondary antibody incubation during 2 h at room temperature. The antibodies used were anti-GFP (1:600, GFP-1020; Aves Labs), anti-RFP (1:1,000, 600-401-379; Rockland antibodies), anti-Nrg1 Type III (1:200, ANR-113; Alomone Labs), anti-GAP43 IgY (1:200, #GAP43; Aves Labs). The secondary antibodies used were anti-chicken 488 (1:500, A-11039; Thermo Fisher Scientific), anti-mouse 488 (1:500, A-21202; Thermo Fisher Scientific), anti-rabbit 555 (1:500, A-31572; Life Technologies), anti-chicken 647 (1:500, N0701-AF647-S; NanoTag Biotechnologies). After staining, the coverslips were mounted in Mowiol for imaging. Fluorescence imaging was performed using an EC PlnN 20x/0.5 DICII objective in a ZEISS observer Z1 with an AxioCam MRm camera and a Colibri 7 laser microscope. We used Fiji/ImageJ software to perform the neurite morphology analysis.

### Dye tracing of the callosal projections and histological procedure

For callosal projection tracing in P0 mice, Nes-Cre animals were perfused with 4% PFA in PBS, and their brains were dissected and post-fixed overnight. Small DiI (1,1′-dioctadecyl-3,3,3′,3′-tetra-methylindocarbocyanine perchlorate) (D3911; Invitrogen) and DiD (1,1'-dioctadecyl-3,3,3',3'-tetramethylindodicarbocyanine) (D7757; Invitrogen) crystals were inserted into the somatosensory cortices of both hemispheres, under a stereo fluorescence microscope (MZ10 F; Leica), as previously described (López-Bendito et al, 2006, 2007). The dye was allowed to diffuse at 37°C in PFA solution for 2 wk. Vibratome sections (80 μm thick) were obtained and counterstained with the fluorescent nuclear dye DAPI (D9542-10MG; Sigma-Aldrich). Slices were imaged within the next two days, to avoid greater crystal diffusion and so being able to identify isolated axons. Fluorescence imaging was performed using a fluorescent microscope Leica DM5000B. We analyzed the sections in the regions of the somatosensory cortex that was identified using the Allen Brain Atlas at P1, Nissl, coronal sections, section #110 (https://developingmouse.brain-map.org/static/atlas), as a reference for the rostro-caudal level. Fiji/ImageJ software was used to perform the analysis.

### IUE and histological procedure

IUE was performed as previously described (Gil-Sanz et al, 2013; Mateos-White et al, 2020). Briefly, E15 C57BL/6J pregnant mice were anesthetized with isoflurane and analgesic solution was injected subcutaneously. The abdominal region was shaved, and after performing an abdominal incision, the two uterine horns were exposed. 0.5–2 μl of the particular endotoxin-free plasmid DNA solution (pCAG-IRES-GFP [PCIG], PCIG + pCAG-Nrg1-FL, or PCIG + pCAG-Nrg1-ICD) was injected into one of the embryos' lateral ventricles. The electroporation conditions involved 5 pulses of 45 V of 80 ms, spaced 950 ms using forceps electrodes (Platinum Tweezertrode, 5 mm Diameter; BTX) and a square wave electroporator (ECM 830 Square Wave Electroporation System; BTX). After surgery, the uterus was returned to the abdominal cavity, and the abdominal wall and skin were sutured. Pregnant females were allowed to give birth. 2-d-old mice (P2) were transcardially perfused with PBS followed by freshly prepared 4% PFA in PBS as previously described (Mateos-White et al, 2020). The brains were cut with a vibratome in 100-μm sections, being ordered in series and processed in floating sections. Primary and secondary antibodies used were diluted in PBS with 0.25% Triton and 4% BSA. Incubation with the primary antibody lasted for 48 h, whereas the secondary antibodies were left overnight at 4°C, both in agitation conditions. The antibodies and dilution used were as follows: anti-GFP 1/500 (GFP-1010; Aves Labs), anti-RFP 1/300 (600-401-379; Rockland antibody), anti-chicken-488 1/500 (A-11039; Thermo Fisher Scientific Scientific), anti-mouse-555 1/500 (A-31570; Invitrogen). Images were taken with a ZEISS observer Z1 with an AxioCam MRm camera and a Colibri 7 laser microscope and a Pln 10x/0.25 Ph1 objective. We analyzed the sections in the regions of the somatosensory cortex that was identified using the Allen Brain Atlas at P1, Nissl, coronal sections, section #110 (https://developingmouse.brain-map.org/static/atlas), as a reference for the rostro-caudal level. At this rostro-caudal level, callosal axons grow parallel to the orientation of the section. The imaging analysis was carried out with Fiji/ImageJ software.

### Imaging

For the in vitro experiments, pictures were taken with a ZEISS observer Z1 microscope equipped with an EC PlnN 20x/0.5 DICII objective, an AxioCam MRm camera, and a Colibri 7 laser system. Images were taken in blind and under the same exposure conditions within the same experiment. ~10 photos per coverslip were taken. For the GAP43 axonal length rescue experiments, an EC PlnN 20x/0.5 DICII objective was used to perform 5 × 5 fields, choosing four random fields per coverslip to have a whole sample representation. The images of the dye tracing were acquired with a Leica DM5000B microscope with N PLAN 2.5x/0.07 and HCX PL Fluotar 10x/0.3 Ph1 objectives, within 2 d after immunofluorescence. To facilitate the analysis, photos were made under two exposure conditions and at two different magnifications: 2.5x for a slice overview and 10x for a better resolution of the callosal projections. Approximately four sections per brain had an optimum dye signal. For the IUE samples, all sections with fluorescence signal were scanned using a ZEISS observer Z1 with an AxioCam MRm camera and a

Colibri 7 laser with an A Pln 10x/0.25 Ph1 objective, generating tiles of the whole corpus callosum and stitching the images with ZEISS ZEN Blue Microscopy software. In addition, we generated, with the EC PlnN 20x/0.5 DICII objective, 17 slices/1.4 interval Z-stacks for the visualization of the more advanced cortico-cortical projecting axons (top axons). In all cases, images were processed and analyzed with Fiji/ImageJ and mounted with Inkscape software. GraphPad was used for statistical analysis.

### Analysis

The in vitro results, both the Nrg1 gain of function and loss of function in primary neuronal cultures, were analyzed using Fiji/ImageJ software and its plugin NeuronJ. First, images were processed by subtracting background (rolling ball radius: 50 pixels) and setting manually a threshold until soma signal saturation. Then, we used NeuronJ to track the longest and thinnest neurite. For the GAP43 axonal length rescue experiments, neurons were randomly chosen in each 5 × 5 field. Then, images were processed by subtracting background (rolling ball radius: 25 pixels) and automatically modified until soma signal saturation. We used the NeuronJ plugin to track the length of the longest and thinnest neurite. For the in vivo analysis of the dye tracing of callosal axons in the Nrg1 KO mouse model, we used Fiji/ImageJ software. All the analysis was carried out in duplicate because of the crystal insertion in the two somatosensory cortices of each brain. We considered the axonal front within the corpus callosum as the most advanced bundle of non-individualized axons. We measured the distance between the midline, as an external non-variable reference, and the front. As a control for the intrinsic variability of the technique, we measured the distance between the insertion zone and the front (data not shown). In addition, the corpus callosum thickness was measured to assess the possibility of major impairments at earlier developmental stages. As another variability control, we also analyzed the area of the crystal insertion zone and its correlation with the observed phenotype. For the in vivo analysis of our Nrg1 gain-of-function model, we analyzed the IUE images using Fiji/ImageJ software. We measured the distance between the electroporation zone and the top 10 more advanced axons in the corpus callosum. This parameter allowed to reduce some of the variability intrinsic to the technique. To confirm the results, we measured the electroporation area and we did a correlation analysis with our phenotype (data not shown). In all the in vivo experiments, we classified the sections using the DAPI staining and the Allen Brain Atlas: Developing Mouse Brain P1 ([http://developingmouse.brain-map.org/experiment/siv?id=100102322&imageId=101634745&initImage=nissl](http://developingmouse.brain-map.org/experiment/siv?id=100102322&imageId=101634745&initImage=nissl)). We compared only analogous regions of different brains.

### Statistical analysis

All statistics were performed with GraphPad Prism software. The details of the statistical methodology can be found in the Statistical Report Table (Table S1). All bar graphs show the mean ± SEM. Significance is indicated with asterisks in each graph, being a *P*-value > 0.05 considered as not significant (ns). The statistical test used for each analysis is mentioned in all figure legends. We used

the Grubbs test with an alpha of 0.01 in GraphPad Prism to identify outliers. For the neuronal culture results, named the gain and loss of function in vitro, an unpaired *t* test was used for comparing Ctrl versus Nrg1 depletion. Conversely, a one-way ANOVA followed by Tukey's multiple comparisons test was used to compare Ctrl versus Nrg1-FL and Nrg1-ICD overexpression. Individual neurons were considered as the sample size. For the protein expression analysis, we used an unpaired *t* test to compare the expression in both Ctrl and Nrg1 KO mice, being independent animals considered as the N. For the loss of function in vivo, we used an unpaired *t* test in the dye-tracing samples and in the corpus callosum thickness analysis, being compared Ctrl and Nrg1 KO mice and taking as the N the different brain sections. In addition, a Pearson correlation coefficient was computed to assess the relation between Nrg1 KO and control neurons. Finally, for the gain of function in vivo, we carried out normality tests in the IUE samples, both for Ctrl versus Nrg1-FL and for Ctrl versus Nrg1-ICD. As the Ctrl versus Nrg1-FL group did not pass the normality test, we compared the conditions using a Kolmogórov–Smirnov test. On the opposite, because of the positive normality test, we used an unpaired *t* test to compare the Ctrl versus Nrg1-ICD conditions. The individual axons were considered as N.

## Data Availability

All reagents and additional information about the results and methodology are available upon request.

## Supplementary Information

## Acknowledgements

We would like to thank all the member of the Lab of Cortical Circuits in Health and Disease and of the Lab of Neuronal Regeneration of CIPF for their feedback, thoughtful discussions, and support during the development of the project. We also thank Alfredo Collado for helping in resource and material supply and Alicia García Jareño for the feedback on expression analysis. I Mateos-White was funded by a Garantía Juvenil contract from the Conselleria de Educación de Valencia (GJIDI/2018/A/221). C Gil-Sanz received a Ramón y Cajal Grant from the Spanish Ministry of Science and Innovation (RyC-2015-19058). This research was partially funded by the Spanish Ministry of Economy and Competitiveness grants SAF-2017-82880-R and PID2020-114227RB-I00 to C Gil-Sanz. This research was funded by the Spanish Ministry of Science and Innovation RYC-2014-16410 and SAF2017-89020-R (to P Fazzari) and PID2021-127112NB-I00 funded by MCIN/AEI/10.13039/501100011033/ and by ERDF A way to make Europe (to G López-Bendito), Generalitat Valenciana (GVA) fellowship ACIF/2019/015 (to A González-Manteiga), Fondos FEDERER to the CIPF from European Community, PROMETEO/2021/052 grant from the Generalitat Valenciana, Conselleria d'Educació, Universitats, i Ocupació (to G López-Bendito as main PI and P Fazzari as Contributing PI), and the Grant PID2020-119779RB-I00 funded by MICIU/AEI/ 10.13039/501100011033 and by ERDF A way to make Europe (to P Fazzari) from Spanish MINECO. PhD funding PRE2018-083562 was granted to A Rodríguez-Prieto.

## Author Contributions

A Rodríguez-Prieto: conceptualization, data curation, formal analysis, validation, investigation, methodology, and writing—original draft, review, and editing.

I Mateos-White: investigation and methodology.

M Aníbal-Martínez: investigation and methodology.

C Navarro-González: investigation and methodology.

C Gil-Sanz: conceptualization, resources, investigation, and methodology.

Y Domínguez-Canterla: investigation and methodology.

A González-Manteiga: conceptualization, formal analysis, and investigation.

V Del Buey Furió: investigation and methodology.

G López-Bendito: conceptualization, resources, supervision, funding acquisition, investigation, and methodology.

P Fazzari: conceptualization, resources, data curation, formal analysis, supervision, funding acquisition, project administration, and writing—original draft, review, and editing.

## Conflict of Interest Statement

The authors declare that they have no conflict of interest.

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
