## [Reviewer comments · Life Science Alliance]

Life Science Alliance

Nrg1 intracellular signaling regulates the development of interhemispheric callosal axons in mice

Angela Rodríguez-Prieto, Isabel Mateos-White, Mar Aníbal-Martínez, Carmen Navarro-González, Cristina Gil-Sanz, Yaiza Domínguez-Canterla, Ana González-Manteiga, Verónica Del Buey Furió, Guillermina Lopez-Bendito, and Pietro Fazzari
DOI: <https://doi.org/10.26508/lsa.202302250>

Corresponding author(s): *Pietro Fazzari, Centro de Investigacion Principe Felipe*

Review Timeline:

Submission Date:	2023-07-04
Editorial Decision:	2023-09-06
Revision Received:	2024-06-06
Editorial Decision:	2024-06-11
Revision Received:	2024-06-14
Accepted:	2024-06-14

Scientific Editor: *Eric Sawey, PhD*

Transaction Report:

September 6, 2023

Re: Life Science Alliance manuscript #LSA-2023-02250-T

Dr. Pietro Fazzari
Centro de Investigacion Principe Felipe
Lab of Cortical Circuits in Health and Disease
Centro de Investigacion Principe Felipe, I64, c/ Eduardo Primo Yufero, 3
Valencia 46012
Spain

Dear Dr. Fazzari,

Thank you for submitting your manuscript entitled "Nrg1 intracellular signaling regulates the development of interhemispheric callosal connections" to Life Science Alliance. The manuscript was assessed by expert reviewers, whose comments are appended to this letter. We invite you to submit a revised manuscript addressing the Reviewer comments.

Thank you for this interesting contribution to Life Science Alliance. We are looking forward to receiving your revised manuscript.

Sincerely,

B. MANUSCRIPT ORGANIZATION AND FORMATTING:

Reviewer #1 (Comments to the Authors (Required)):

The manuscript entitled "Nrg1 intracellular signaling regulates the development of interhemispheric callosal connections" (please add mention of the animal specie in title: "in mice") by Angela Rodriguez-Prieto and collaborators explores the roles of Nrg1 in glutamatergic cortical neurons development. Specifically, the authors demonstrate that loss of Nrg1 leads to reduced axon length in vivo (through tracer injection) and in vitro, whereas Nrg1 overexpression increases axon length in vitro and in vivo (through in utero cortical electroporations). Interestingly, this effect can be observed as well by expressing the cleaved intracellular domain of Nrg1, providing some mechanistic insight.

The data is clear and conclusions are supported by the experiments. The advance from this paper is not major, since other papers demonstrate that Nrg1 affects axon length in gabaergic neurons (as discussed by the authors), but also in excitatory neurons (see for example Zhang et al. Sci. Rep. 2017, doi: 10.1038/srep42525). Still, this study has value to the community and I would support publication of a revised version. Below are suggestions that would reinforce the impact of the paper.

1. On a mechanistic point of view, the authors suggest that Nrg1 cleavage and release of the ICD peptide is tied to the axon elongation phenotype (Figures 3 and 4). This could be further demonstrated by blocking Nrg1 cleavage either with a g-secretase inhibitor, or by expressing a Nrg1 mutant which cannot be cleaved (if this exists).
2. Quantification of axon length could be completed with quantifications of axon ramification (number of branchpoints) as well as dendritic morphology to describe if the phenotype is axon specific or if the whole neuronal morphology is affected.
3. The observation of a growth phenotype at P2 is intriguing, but raises the question if accelerated growth has some consequences on cortical circuits building. Would axons growing faster continue to develop abnormal connections? Reach the wrong targets? or rather, would they prune because they reach targets before the local environment is favourable? Looking at a later timepoint (eg. P30) where axonal connections are fully developed would provide much information.

Technical comments

1. Please provide graphical representation that show individual datapoints so that the reader can appreciate experimental variability. Histograms and error bars represent a loss of information for the reader.
2. Figure 1: It is hard to visualize axon development from the highly truncated pictures, especially since there is no image showing brain slice morphology (aside from a cartoon). This information is important for the reader to demonstrate that images are taken at the same brain level (as should be for stereotactic injections) and that the orientation of the cut is perfectly transversal. Any shift in the rostro-caudal axis or laterally could provide an effect mimicking changes in axon growth. Similarly, since on average 4-5 slices were collected per brain, what is the variability within one single animal? What are the differences of position of the slice on a rostro-caudal axis? One might advise to show and quantify only one slice (always the same position) per animal, if there's a robust effect of reduction on axon length then it should be evident as well.
3. Figure 4: similarly to Figure 1, it would be better for the reader to present a picture of the entire slice and to explain if the sections chosen for the quantification have been taken at the same position on a rostro-caudal axis. Once again, changes in the cutting plan can have artificial impact on the quantification.
4. Quantifications of axon length in vitro from figures 2 and 3 should be presented as absolute values in um rather than normalized values. If normalization was rendered necessary because of culture to culture variability, the raw data can be presented in supplementary. Still this information is important to the reader to compare to the literature and estimate the viability of neuronal cultures. Specifically, from the scale bar and the pictures and traces in figure 3, one might think that Ctrl neurons in figure 3 are actually less developed than the KO neurons in figure 2. If so, one might overestimate the magnitude of the axon elongation phenotype upon overexpression of Nrg1 and Nrg1-ICD.
5. Figure 4 paned D: it seems that axons are fragmented in both the control and ICD conditions. Is this an artefact due to tissue fixation/preparation? If so, please provide a better picture. Otherwise, it could be that on this batch of electroporation there has been some sort of neurotoxic effect of plasmid expression, which could affect axon growth and decreases confidence in the result.

Reviewer #2 (Comments to the Authors (Required)):

The major risk factor for schizophrenia, Neuregulin 1 (NRG1), plays a crucial role in myelination, neurite growth, and spine formation. Here, the authors investigated the influence of NRG1 on callosal axon length and long-range cortico-cortical connectivity in relation to schizophrenia. The study reveals that Nrg1 deficiency affects callosal axon development in the mouse brain. Similarly, Nrg1-deficient primary neurons exhibit shortened axons. Conversely, gain-of-function experiments using Nrg1-FL and Nrg1-ICD overexpression clearly display heightened axon length in cultured neurons and in the brain.

Data presented in manuscript supports the conclusions drawn by authors. While there is potentially new and useful information here and the results appear interesting, the authors presentation lacks details and additional evaluations to validate their experimental models. The presentation of the figures could be better, and the authors need to discuss functional and behavioral aspects of the data and its interpretation. A few concerns exist:

Major:

1. Schizophrenic patients exhibit cognitive impairments and alter social behavior. The behavioral phenotype of Nrg1-deficient mice should be discussed in this context.
2. Gender-based differences have been reported in patients with schizophrenia on onset, symptoms, and social behavior. The present study does not clearly state the exact number or ratio of male and female mice used. What was the rationale for such a selection? And were there any differences in results between male and female mice?
3. The functional aspect of the connectivity deficit in relation to the pathophysiology of schizophrenia should be discussed in detail.
4. For the characterization of in vivo NRG1 deletion, authors should confirm change at the protein level using staining, blotting, or similar techniques. Immunostaining data could be useful to understand region- and cell-specific changes in NRG1 levels in the brain.
5. Similar to above, what is the NRG1 protein level in Nrg1 KO and Nrg1-FL/Nrg1-ICD overexpressing neurons compared to control neurons? Western blot data will be useful in this context.
6. Since overexpression of Nrg1-FL/Nrg1-ICD not only restores axon length but can also increase axon length more than that of control neurons, Is there any correlation between NRG1 protein level and axon length in cultured neurons?
7. The adverse effects of NRG1 overexpression should also be added to the discussion.

Minor:

1. Graphs should show individual data points and bidirectional error bar.
2. What is the rational of using relative ratio in figure 2B, 3C, 4B and 4E rather than raw reading?
3. Please describe the method used to identify outliers in data?
4. In figure legend of supplementary figure 1B. it mentioned "Ctrl = 3 and Nrg1 KO = 2, littermates"; please explain the n number. Did authors used only two Nrg1 KO pups to obtain the data in supplementary Figure 1B and draw conclusions?
5. For all data sets please provide full statistical reports.
6. The method section describes mRNA isolation from cultured neurons, but there are no related data.

Reviewer #3 (Comments to the Authors (Required)):

The article titled "Nrg1 intracellular signaling regulates the development of interhemispheric callosal connections" explores the role of the schizophrenia (SZ) risk gene Nrg1 in the development of interhemispheric callosal connections. The authors demonstrate a cell-autonomous function of Nrg1 in excitatory neurons, where it influences axon growth in the corpus callosum. The findings reveal that Nrg1 deletion leads to underdeveloped axons, while overexpression of Nrg1 results in increased axon growth. This effect is attributed to the activation of Nrg1, leading to the release of the intracellular domain (ICD), which in turn promotes axon growth. The study presents high-quality data and effectively presents its results. However, certain concerns warrant attention:

1. The study highlights that Nrg1 knockout (KO) brains exhibit underdeveloped callosal axons and hypoconnectivity, yet there is no observed difference in the size of the corpus callosum. This observation contrasts with reports associating reduced corpus callosum size with SZ patients. It would be valuable to discuss how the hypoconnected corpus callosum maintains its size in Nrg1-null brains.
2. Dye tracking experiments were conducted on postnatal day 0 (P0) animals. It would be informative to extend the analysis to assess the state of these axons at postnatal day 30 (P30) to gain insight into the long-term effects of Nrg1 manipulation.
3. Several studies have linked SZ with the overexpression of Nrg1, such as the work by Olaya et al., 2017, which reported Nrg1 type III overexpression in the brain leading to SZ-like behavior. Considering this, it is important to address the association between loss of function, axon undergrowth, and SZ. How does the observed axon undergrowth in the context of Nrg1 deletion

relate to the reported Nrg1 overexpression and its behavioral consequences in SZ?

4. Rescue experiments on Nrg1 KO cultures by transfecting with full-length Nrg1 and ICD versions, could provide insights into whether axon growth can be rescued to wild-type levels.

5. Given the conflicting reports regarding the role of Nrg1 in SZ, it would be valuable to explore the extent of Nrg1 expression in cells transfected with Nrg1 full-length and Nrg1-ICD variants. Additionally, investigating the degree of overexpression associated with axon overgrowth and its correlation with SZ-like behavior could provide a clearer understanding of the complex relationship between Nrg1 expression levels and SZ-associated phenotypes.

REBUTTAL LETTER

Point-to-point Rebuttal letter for the Manuscript # LSA-2023-02250-TR – “*Nrg1 intracellular signaling regulates the development of interhemispheric callosal connections*” by Rodriguez-Prieto et al.

To facilitate the work of the Editors and Reviewers, we used the following formats to distinguish between Reviewer’s comments, our response, and the text amended in the manuscript.

Formatting code:

Arial Blue, Reviewers feedback

Arial Black, our Response

Times new Roman Black, original the text of the Manuscript,

Times new Roman Green, changes to the original the text of the Manuscript

The position of the changes is referenced using the revised text (e.g. page 8, line 235).

Reviewer #1 (Comments to the Authors (Required)):

General response to all Reviewers

We would like to thank all the three Reviewers for their effort and for their very positive and constructive feedback. We particularly appreciate that all three Reviewers agree that the evidence presented in the manuscript is sound and robustly supports the conclusions, indicating the value of our study to the community. For instance: Reviewer #1, “The data is clear and conclusions are supported by the experiments”; Reviewer #2, “Data presented in manuscript supports the conclusions drawn by authors”; Reviewer #3, “The study presents high-quality data and effectively presents its results”.

As it is usually the case when multiple reviewers provide feedback on a manuscript, the feedback from the three Reviewers provided different perspectives and directions for investigation. As we understand it, the overall suggestions mainly encouraged us to provide further mechanistic insights into the role of Nrg1 in axonal development and to explore a possible role of Nrg1 at the adult stage.

We have significantly improved the manuscript by addressing the Reviewers' feedback. Specifically, to further improve the quality of our study, we i) identified GAP43 as a relevant effector of Nrg1 signaling, ii) deepened the analysis of the cellular effects of Nrg1 signaling in neurite outgrowth, and iii) explored the role of Nrg1 in maintaining callosal projections through loss-of-function experiments in the cortices of adult mice. In addition to these main points, we have responded extensively to the reviewers' input, as detailed point-by-point below.

Point-to-point rebuttal to Reviewer #1

Reviewer #1

The manuscript entitled “Nrg1 intracellular signaling regulates the development of interhemispheric callosal connections” (please add mention of the animal specie in title: “in mice”) by Angela Rodriguez-Prieto and collaborators explores the roles of Nrg1 in glutamatergic cortical neurons development. Specifically, the authors demonstrate that loss of Nrg1 leads to reduced axon length in vivo (through tracer injection) and in vitro, whereas Nrg1 overexpression increases axon length in vitro and in vivo (through in utero cortical electroporations). Interestingly, this effect can be observed as well by expressing the cleaved intracellular domain of Nrg1, providing some mechanistic insight.

The data is clear and conclusions are supported by the experiments. The advance from this paper is not major, since other papers demonstrate that Nrg1 affects axon length in gabaergic neurons (as discussed by the authors), but also in excitatory neurons (see for example Zhang et al. *Sci. Rep.* 2017, doi: 10.1038/srep42525). Still, this study has value to the community and I would support publication of a revised version. Below are suggestions that would reinforce the impact of the paper.

Reply:

We thank Reviewer # 1 for taking the time to evaluate our manuscript. We are particularly thankful for appreciating that our results are clear and support the conclusion of our study. We believe that this is particularly important in a complex study that presents both *in vivo* and *in vitro* approaches with mechanistic insights.

With regard to the novelty of the study, we thank the author for acknowledging that “this study has value to the community”. This feedback also stimulated us to improve the “Introduction” with regard to the novelty of our study in the context of the previous literature. Specifically:

Regarding the comment “add mention of the animal specie”, we amended the Title as requested.

New title: "Nrg1 intracellular signaling regulates the development of interhemispheric callosal connections in mice"

Concerning the comment, “Nrg1 affects axon length in gabaergic neurons (as discussed by the authors)”, we amended the text in the introduction to explain more clearly our previous findings on the role of Erbb4 activation inhibitory neurons. To facilitate the work of Reviewer #1 we paste here the amended paragraph that can be found in the revised Manuscript. In Green we highlight the amended Text.

See Introduction, page 4 line 5

“These studies demonstrated that Nrg1/Erbb4 signaling plays an important role in the cortex, and specifically in the wiring of inhibitory cortical neurons that express the Nrg1 receptor Erbb4 (Bjarnadottir et al, 2007; Li et al, 2007; Mei & Xiong, 2008; Chen et al, 2010a, 2010b; Fazzari et al, 2010, 2014; Pedrique & Fazzari, 2010; Rahman-Enyart et al, 2020; Navarro-Gonzalez et al, 2021). Erbb4 activation in inhibitory neurons is required for proper wiring of local inhibitory circuits, as it promotes the growth of inhibitory axons in vitro the formation of GABAergic synapses in vitro and in vivo (Fazzari et al, 2010; Rico & Marín, 2011; Navarro-Gonzalez et al, 2021).”

Regarding the role of Nrg1 in excitatory neurons, Reviewer #1 cited the manuscript of Zhang et al. Sci. Rep. 2017, doi: 10.1038/srep42525. We thank Reviewer #1 for mentioning this study. We did not include earlier this reference because we considered it redundant with Chen, Y.,... Talmage, D. A; *J. Neurosci.* 30, 9199-9208, which we have cited instead in both the Introduction and Discussion. The paper by Chen et al. in 2010 was, to our knowledge, the first to extensively investigate the role of Nrg1 in excitatory neurons. Moreover, we understand that Zhang et al. (2017) primarily focused on dendrites, based on their use of the dendritic marker MAP2 and the presented images. Notably, Zhang et al. [*Sci. Rep.* 2017, doi: 10.1038/srep42525] follow up on a previous manuscript from the same group, namely Zhang et al., *Sci Rep.* 2016. doi: 10.1038/srep19581. Zhang et al. 2016 present the analysis of neurite outgrowth in NRG1 KO primary neurons using the same approach (MAP2 labeling). Despite the fact that, to our understanding, Zhang et al. (2016 and 2017) focused on dendrites, for the sake of completeness we have now included their references in the amended manuscript.

See Discussion, page 12 line 8:

“The role of Nrg1 in axonal development in pyramidal neurons is poorly understood. Nonetheless, a few studies suggested that Nrg1 loss-of-function may impair dendritic development (Zhang et al, 2016, 2017).”

Reviewer #1:

1. On a mechanistic point of view, the authors suggest that Nrg1 cleavage and release of the ICD peptide is tied to the axon elongation phenotype (Figures 3 and 4). This could be further demonstrated by blocking Nrg1 cleavage either with a g-secretase inhibitor, or by expressing a Nrg1 mutant which cannot be cleaved (if this exists).

Reply:

This is a sensible suggestion. Indeed, previous studies have demonstrated that inhibiting gamma-secretase activity leads to a decrease in neurite outgrowth (see for instance Barao S. and De Strooper, 2016; Javier-Torrent et al., 2019; Nathalie Jurisch-Yaksi et al., 2013). While this finding supports our hypothesis regarding the involvement of Nrg1 cleavage in axonal development, caution must be exercised when employing gamma-secretase inhibitors as a tool. The primary concern lies in the broad substrate specificity of gamma-secretase, which processes over 50 proteins. Consequently, determining the optimal concentration of a gamma-secretase inhibitor may prove challenging; a low concentration might yield no discernible effect, while a high concentration could induce toxicity unrelated to Nrg1 processing, thereby impairing neuronal survival. Regarding the use of an uncleavable Nrg1, the Nrg1 V321L mutant reported by Dejaegere et al. (2008) would be a possible option. However, this mutation only partially reduces Nrg1 processing, making this option less straightforward.

In summary, while the reviewer's suggestion is quite reasonable, its implementation is not straightforward due to the variety of potential gamma-secretase targets implicated in neurite growth and survival, in addition to Nrg1. Overall, we believe that the expression of Nrg1-ICD represents the most specific experiment to elucidate the role of Nrg1 intracellular signaling, as demonstrated by our work and that of others (Fazzari, 2014; Navarro-González, 2019; Bao, 2003).

Reviewer

#1:

2. Quantification of axon length could be completed with quantifications of axon ramification (number of branchpoints) as well as dendritic morphology to describe if the phenotype is axon specific or if the whole neuronal morphology is affected.

Reply:

We appreciate the reviewer's suggestion to further strengthen the cellular analysis of the role of Nrg1 signaling in neurite outgrowth. In response to reviewer #1's constructive input, we have expanded our study to include quantification of axonal branching patterns (number of branch points per length) in all *in vitro* experiments. The additional data are shown in the Main Figures. We also performed a Sholl analysis of dendritic arborization in all experimental conditions. The quantifications of the Sholl analysis are shown in Supplementary Figures.

Reviewer

#1:

3. The observation of a growth phenotype at P2 is intriguing, but raises the question if accelerated growth has some consequences on cortical circuits building. Would axons growing faster continue to develop abnormal connections? Reach the wrong targets? or rather, would they prune because they reach targets before the local environment is favourable? Looking at a later timepoint (eg. P30) where axonal connections are fully developed would provide much information.

Reply:

Reviewer #1 raises an interesting point, which is certainly worthy of future investigation. Callosal development is a complex, multi-step process. As the reviewer points out, our

observation that Nrg1 expression by IUE accelerates callosal axon growth doesn't necessarily translate into improved interhemispheric connectivity at later stages. We agree that Nrg1-induced accelerated growth may be compensated for later by pruning and target refinement. Alternatively, Nrg1-expressing neurons might overshoot and reach inappropriate targets.

Our current study focuses primarily on the early stages of callosal development and does not address the role of Nrg1 in later stages of interhemispheric connectivity. However, we appreciate the suggestion to explore the function of Nrg1 at later stages, which could enrich our understanding of callosal circuitry. Therefore, we undertook the following experimental approaches to address this suggestion:

- First, we performed IUE experiments to assess the postnatal effects of Nrg1 expression. Unfortunately, we were unable to obtain sufficient numbers of electroporated mice due to maternal infanticide in mothers experiencing perinatal stress. This is a recognized problem in this experimental paradigm and we were unable to overcome it during the revision process.

- Second, we investigated whether Nrg1 signaling is necessary for maintaining callosal projection integrity in adults by using inducible UBC-CreER2 mice for targeted Nrg1 deletion. Our findings, presented in Supplementary Figure 6, indicate no significant differences in the profile of the callosal connections, suggesting that Nrg1 may play a redundant role once callosal projections are established. However, it remains possible that Nrg1 influences synaptic plasticity or neurotransmitter release in callosal neurons at this stage. This hypothesis would be consistent with previous studies from our lab and others regarding Nrg1's role in cortical wiring and synaptic transmission (see references in the manuscript).

Future studies, as suggested in the manuscript (page 9, line 27), could explore these possibilities using synaptic and cellular markers to investigate the contributions of Nrg1 to interhemispheric synaptic wiring.

Reviewer #1

Technical comments

1. Please provide graphical representation that show individual datapoints so that the reader can appreciate experimental variability. Histograms and error bars represent a loss of information for the reader.

Reply:

We modified all the graphs to include the individual datapoints as suggested.

Reviewer #1

2. Figure 1: It is hard to visualize axon development from the highly truncated pictures, especially since there is no image showing brain slice morphology (aside from a cartoon). This information is important for the reader to demonstrate that images are taken at the same brain level (as should be for stereotactic injections) and that the orientation of the cut is perfectly transversal. Any shift in the rostro-caudal axis or laterally could provide an effect mimicking changes in axon growth. Similarly, since on average 4-5 slices were collected per brain, what is the variability within one single animal? What are the differences of position of the slice on a rostro-caudal axis? One might advise to show and quantify only one slice (always the same position) per animal, if there's a robust effect of reduction on axon length then it should be evident as well.

Reply:

We appreciate the critical input from reviewer #1. Indeed, the *in vivo* tracing experiments shown in Figures 1 and 4 are complex and require precise execution. We were fortunate to collaborate with two highly specialized laboratories: the lab of Guillermina Lopez-Bendito for the dye tracing in Figure 1 (e.g. Moreno-Juan et al., Nat Comm, 2017; Antón-Bolaños N. et al.,

Science, 2019) and the lab of Cristina Gil-Sanz for in utero electroporation (e.g. Fabra-Beser J et al., J Neuro, 2021; Gil-Sanz C. et al., Neuron, 2015). We took great care with technical aspects such as brain orientation and included only consistently labeled and processed specimens in our analysis.

In response to the reviewer's suggestions, we have improved the quality of Figure 1. We have selected new representative images for control and Nrg1 KO mice and included DAPI staining (Figure 1). In addition, we have added a low magnification image in the Supplementary Material to show the entire section as suggested (Supplemental Figure 1D).

Regarding the number of sections and the suggestion to quantify only one section, we thank reviewer #1 for prompting us to clarify our methodology. We sectioned and photographed the entire brain, but performed the analysis at a specific and limited rostrocaudal level in the somatosensory cortex. We used the Allen Brain Atlas at P1, Nissl, coronal sections, section #110 (<https://developingmouse.brain-map.org/static/atlas>) as a reference, which we have now added to the Materials and Methods section. Because we labeled both sides of the brain with different dyes (as described in the Methods section), we obtained two measurements per section, which increases the robustness of our analysis. Although rostro-caudal diffusion of the dye may vary, this approach allowed us to obtain an average of six measurements from three consecutive sections. Each section was 80 μm thick, covering a total of 240 μm , a relatively small span, resulting in very consistent data.

We believe these revisions effectively address Reviewer #1's concerns and significantly improve the clarity and strength of Figure 1.

Reviewer #1

3. Figure 4: similarly to Figure 1, it would be better for the reader to present a picture of the entire slice and to explain if the sections chosen for the quantification have been taken at the same position on a rostro-caudal axis. Once again, changes in the cutting plan can have artificial impact on the quantification.

Reply:

Similar to the previous point, axonal growth was consistently quantified at the rostro-caudal level of the somatosensory cortex using the reference Allen Brain Atlas at P1, Nissl, coronal sections, section #110 (<https://developingmouse.brain-map.org/static/atlas>). We focused on this region because, at this rostro-caudal level, callosal axons grow parallel to the orientation of the section at this stage. Therefore, quantification of axonal elongation at this rostro-caudal level is straightforward and robust. We have also included a low magnification image of the sections with DAPI counterstaining in Supplementary Figure 5C,D as suggested.

Reviewer #1

4. Quantifications of axon length in vitro from figures 2 and 3 should be presented as absolute values in μm rather than normalized values. If normalization was rendered necessary because of culture to culture variability, the raw data can be presented in supplementary. Still this information is important to the reader to compare to the literature and estimate the viability of neuronal cultures. Specifically, from the scale bar and the pictures and traces in figure 3, one might think that Ctrl neurons in figure 3 are actually less developed than the KO neurons in figure 2. If so, one might overestimate the magnitude of the axon elongation phenotype upon overexpression of Nrg1 and Nrg1-ICD.

Reply:

We welcome the opportunity to provide further clarification of our methodology. We agree that presenting axon length quantifications in absolute values (μm) may provide additional context for the reader to compare with the existing literature. To address this concern, we have included the absolute raw values of axon length in Supplementary Figures (FS2, FS3, FS4). Statistical analysis

of the non-normalized raw data strongly confirms the significance of the differences between experimental conditions and controls, further strengthening the robustness of our results.

We have left the evaluation of fold change relative to internal controls in the main figure because we are convinced that this is a well-established and scientifically sound approach in the field to compensate for normal experimental variability. In this regard, we respectfully invite Reviewer #1 to review our previous publications and those of other reputable labs (such as the labs of Oscar Marin, Beatriz Rico, Bart De Strooper, and Carlos Dotti; see the references in the manuscript for details).

Regarding the observed variability in neuronal development between the control conditions in Figures 2 and 3, it is well within the expected range for this experimental model and does not affect the overall conclusions of our study. Certainly, this small difference is not due to cell viability issues. Indeed, regarding the issue of "the viability of neuronal cultures", we perform stringent quality control measures, including thorough microscopic evaluation on different days of cell density, morphology, and absence of cell debris, to ensure the health and viability of our neuronal cultures. Any cultures showing signs of compromised viability or abnormal development are discarded.

In summary, we have addressed the reviewer's concern by including the absolute values of axon length in microns in the Supplementary Figures. We have also clarified our quality control measures and the rationale for the use of internal controls in this experimental paradigm. Overall, we believe that these changes strengthen the manuscript and provide the reader with a more complete understanding of our findings.

Reviewer #1

5. Figure 4 paned D: it seems that axons are fragmented in both the control and ICD conditions. Is this an artefact due to tissue fixation/preparation? If so, please provide a better picture. Otherwise, it could be that on this batch of electroporation there has been some sort of neurotoxic effect of plasmid expression, which could affect axon growth and decreases confidence in the result.

Reply:

We appreciate Reviewer #1's careful examination of Figure 4D. In our experience working with *in utero electroporation*, which is the expertise of Cristina Gil-Sanz's lab, it is not uncommon to observe some varicosities or irregularities in dendrites and axons at high magnification in these *in vivo* preparations (see for instance Guo et al., Nat Commun 2015, PMID: 26206566; and Fabra-Beser et la., J Neuro 2021, PMID: 34266896). Importantly, these irregularities are seen in 1) top-growing axons, suggesting that they are not related to impaired growth and 2) in both GFP control and Nrg1-expressing neurons, arguing against a neurotoxic effect of Nrg1 expression.

Regarding the quality of the images, while we could enhance the images by saturating them for more visually pleasing results, we prefer to maintain the integrity of the data by presenting images that are as close as possible to the raw data. Therefore, with the kind permission of Reviewer #1 and the editor, we would prefer to keep the images as they are.

Reviewer #2 (Comments to the Authors (Required)):

General response to all Reviewers

We would like to thank all the three Reviewers for their effort and for their very positive and constructive feedback. We particularly appreciate that all three Reviewers agree that the evidence presented in the manuscript is sound and robustly supports the conclusions, indicating the value of our study to the community. For instance: Reviewer #1, “The data is clear and conclusions are supported by the experiments”; Reviewer #2, “Data presented in manuscript supports the conclusions drawn by authors”; Reviewer #3, “The study presents high-quality data and effectively presents its results”.

As it is usually the case when multiple reviewers provide feedback on a manuscript, the feedback from the three Reviewers provided different perspectives and directions for investigation. As we understand it, the overall suggestions mainly encouraged us to provide further mechanistic insights into the role of Nrg1 in axonal development and to explore a possible role of Nrg1 at the adult stage.

We have significantly improved the manuscript by addressing the Reviewers' feedback. Specifically, to further improve the quality of our study, we i) identified GAP43 as a relevant effector of Nrg1 signaling, ii) deepened the analysis of the cellular effects of Nrg1 signaling in neurite outgrowth, and iii) explored the role of Nrg1 in maintaining callosal projections through loss-of-function experiments in the cortices of adult mice. In addition to these main points, we have responded extensively to the reviewers' input, as detailed point-by-point below.

Point-to-point rebuttal to Reviewer #2

The major risk factor for schizophrenia, Neuregulin 1 (NRG1), plays a crucial role in myelination, neurite growth, and spine formation. Here, the authors investigated the influence of NRG1 on callosal axon length and long-range cortico-cortical connectivity in relation to schizophrenia. The study reveals that Nrg1 deficiency affects callosal axon development in the mouse brain. Similarly, Nrg1-deficient primary neurons exhibit shortened axons. Conversely, gain-of-function experiments using Nrg1-FL and Nrg1-ICD overexpression clearly display heightened axon length in cultured neurons and in the brain. Data presented in manuscript supports the conclusions drawn by authors. While there is potentially new and useful information here and the results appear interesting, the authors presentation lacks details and additional evaluations to validate their experimental models. The presentation of the figures could be better, and the authors need to discuss functional and behavioral aspects of the data and its interpretation. A few concerns exist:

Reply:

We thank Reviewer #2 for all the valuable feedback on our manuscript. We appreciate the acknowledgement of the soundness of our findings on the role of Neuregulin 1 (NRG1) and the conclusions we draw from these results. We have improved our discussion to clarify the functional and behavioral significance of NRG1 in neuronal development and schizophrenia. In addition, we have significantly improved the presentation of the figures and further clarified the methodology, including the statistical methodology. We believe that this revision fully addresses the comments of Reviewer #2.

Reviewer #2

Major:

1. Schizophrenic patients exhibit cognitive impairments and alter social behavior. The behavioral phenotype of Nrg1-deficient mice should be discussed in this context.

Reply:

We agree with the reviewer that the behavioral impact of Nrg1 deficiency is relevant. We have revised the text to acknowledge the reviewer's point and provided references to the relevant literature. While our current focus is on cellular and molecular neurobiology, we have included a

reference to key studies demonstrating SZ-like behavioral phenotypes in mice mutant for *Nrg1/Erbb4* forward and intracellular signaling (page 3, line 85). These studies, extensively reviewed by Mei (2008) and Mei & Xiong (2014), provide strong evidence for the link between *Nrg1* signaling and schizophrenia-related behaviors.

Page 3, line 30:

“Numerous studies have identified the *neuregulin 1 (NRG1)* gene as a risk factor for the development of schizophrenia in various populations (Stefansson et al, 2002; Williams et al, 2003; Yang et al, 2003; Shyu et al, 2004; Tang et al, 2004; Harrison & Weinberger, 2005; Mei & Xiong, 2008). Interestingly, several studies in preclinical mouse models have shown that various genetic mutations that impair *Nrg1/Erbb4* forward and intracellular signaling exhibit SZ-like symptoms, such as working memory deficits and hypersensitivity to psychostimulants (Stefansson et al, 2002; Coolen et al, 2005; Dejaegere et al, 2008; Mei & Xiong, 2008; Mei & Nave, 2014).”

Reviewer #2

2. Gender-based differences have been reported in patients with schizophrenia on onset, symptoms, and social behavior. The present study does not clearly state the exact number or ratio of male and female mice used. What was the rationale for such a selection? And were there any differences in results between male and female mice?

Reply:

We appreciate Reviewer #2 raising the important issue of sex differences in schizophrenia. However, our study focused on primary neuronal cultures and neonatal pups. In these experimental paradigms, sexual dimorphism is not well established in mice. To our knowledge, the current scientific literature does not address sex differences in mouse callosal development at these early developmental stages.

For the neuronal cultures, our model uses single-cell transfection, and these cultures are typically established by pooling cortical tissue from multiple embryos without regard to sex. Thus, the potential impact of sex differences is minimized in this context.

We hope this clarifies the rationale behind our experimental design and addresses the reviewer's concern.

Reviewer

#2

3. The functional aspect of the connectivity deficit in relation to the pathophysiology of schizophrenia should be discussed in detail.

Reply:

We appreciate the reviewer's suggestion to discuss the functional aspect of the altered connectivity observed in *Nrg1*-deficient mice and its potential implications in schizophrenia. In response, we have revised the Discussion section to more clearly articulate how the alterations in corpus callosum development observed in *Nrg1*-deficient mice may affect interhemispheric connectivity and brain function.

Specifically, in the original manuscript, we devoted four paragraphs of the Discussion (from page 10, line 14 to page 11, line 19) to describing the development of the corpus callosum and its pathophysiological functions in humans. In response to the reviewer's feedback, we have modified this section to more clearly describe how the alterations in corpus callosum development observed in *Nrg1*-deficient mice may affect interhemispheric connectivity and brain function (Discussion, page 10, line 14 to page 11, line 19).

We believe that these revisions significantly improve the clarity and readability of the Discussion and fully address the reviewer's feedback. We thank the reviewer for the valuable input.

Reviewer #2

4. For the characterization of *in vivo* NRG1 deletion, authors should confirm change at the protein level using staining, blotting, or similar techniques. Immunostaining data could be useful to understand region- and cell-specific changes in NRG1 levels in the brain.

Reply:

We thank the reviewer for this suggestion. We would like to emphasize that the Nrg1 flox mouse model is well established and numerous studies over the past 20 years have confirmed that Nrg1 is effectively deleted upon Cre expression (J:69623 Yang X, et al., Patterning of muscle acetylcholine receptor gene expression in the absence of motor innervation. *Neuron*. 2001 May;30(2):399-410 was the first reference. 20 references can be found at <https://www.informatics.jax.org/reference/allele/MGI:2447761?typeFilter=Literature>)

In our study, we further validated this deletion by showing that Nrg1 mRNA expression is abolished using qPCR. Since protein expression is not possible in the absence of mRNA, we believe that our approach provides an accurate and sensitive confirmation of Nrg1 deletion.

In addition, commercially available antibodies against NRG1 are not very reliable. Therefore, we respectfully suggest that a Western blot would not significantly increase the validity of our results compared to the qPCR data we have provided.

Reviewer #2

5. Similar to above, what is the NRG1 protein level in Nrg1 KO and Nrg1-FL/Nrg1-ICD overexpressing neurons compared to control neurons? Western blot data will be useful in this context.

Reviewer #2

6. Since overexpression of Nrg1-FL/Nrg1-ICD not only restores axon length but can also increase axon length more than that of control neurons, Is there any correlation between NRG1 protein level and axon length in cultured neurons?

Reply to points 5 and 6:

We reply together to the reviewer comments 5 and 6 since they are related.

We appreciate the reviewer's valuable suggestions regarding directly measuring Nrg1 protein levels and their potential correlation with axon length.

Our study design utilizes co-culturing with non-electroporated neurons to achieve sparse labeling and single-cell resolution. This approach offers the significant advantage of analyzing Nrg1 manipulation effects at the single-cell level while maintaining an environment with endogenous Nrg1 levels. However, it presents a challenge for Western blot analysis, as it wouldn't definitively distinguish between exogenous and endogenous Nrg1.

As the reviewer rightly pointed out, electroporation inherently leads to variable Nrg1 expression within the cell population, making quantification of this heterogeneity within the entire pool a significant challenge. While a theoretical correlation between Nrg1 levels and axon length may exist, our current quantification methods represent the average for the Nrg1-expressing population, which is a standard caveat of electroporation and transfection experiments.

In summary, while we appreciate the reviewer's suggestions, we acknowledge the technical challenges associated with the proposed Western blot analysis and correlation quantification. We believe that the current data, demonstrating the effects of Nrg1 manipulation on axon length, remains informative and sufficient to support the study's conclusions.

Reviewer #2

7. The adverse effects of NRG1 overexpression should also be added to the discussion.

Reply:

We agree with reviewer #2 and appreciate this constructive suggestion. In response, we have modified the Discussion to include relevant references on this topic and its relevance to inhibitory/excitatory homeostasis in the brain. Specifically, we have added the following paragraph to the Discussion (page 12, line 1):

“Interestingly, while most studies in preclinical models have focused on loss of Nrg1/ErbB4 signaling, others have shown that exogenous expression of Nrg1 can also be detrimental to cortical wiring and lead to SZ-like symptoms (Hahn et al, 2006; Yin et al, 2013; Agarwal et al, 2014; Olaya et al, 2017). These results suggest that an optimal level of Nrg1 is required to maintain homeostasis of excitatory/inhibitory circuits in the cortex (Agarwal et al, 2014).”

Reviewer

#2

Minor:

1. Graphs should show individual data points and bidirectional error bar.

Reply:

The graphs were modified according to the reviewer's suggestion.

Reviewer #2

2. What is the rational of using relative ratio in figure 2B, 3C, 4B and 4E rather than raw reading?

Reply:

We thank Reviewer #2 for this feedback. As mentioned in detail in our response to Reviewer #1 (Technical comment 4), using an internal control as a reference to evaluate the effect of gene expression or deletion is a well-established and commonly used approach, particularly in early neurodevelopmental studies. This method addresses inherent experimental variability and biological noise, especially when working with primary neuronal cultures. Normalizing the data to an internal control enhances the consistency and reliability of the analysis. Additionally, in response to Reviewer #1's specific suggestion, we have included the absolute raw values of axon length for all primary neuronal culture experiments in the Supplementary Figures (SF2, SF3, SF4). This provides additional context for readers who prefer raw data.

Reviewer #2

3. Please describe the method used to identify outliers in data?

Reply:

We thank Reviewer #2 for raising this issue. We used the iterative Grubbs test with an alpha of 0.01 in GraphPad Prism to identify outliers. This test worked well for our data set and demonstrated robustness and consistency in our analysis. We have added the use of the iterative Grubbs test with the alpha value to the Methodology section (page 22, line1). In addition, the results of this outlier analysis are described in the statistical report table as suggested in item 5.

Reviewer #2

4. In figure legend of supplementary figure 1B. it mentioned "Ctrl = 3 and Nrg1 KO = 2, littermates"; please explain the n number. Did authors used only two Nrg1 KO pups to obtain the data in supplementary Figure 1B and draw conclusions?

6. The method section describes mRNA isolation from cultured neurons, but there are no related data.

Reply to points 4 and 6:

We thank reviewer #2 for pointing out these issues. We acknowledge an editing error in Supplementary Figure 1B and apologize for the confusion.

The nestin Cre-deleter drives Cre expression as early as embryonic day 11. This line has been validated in many publications and we have carefully verified that it works as expected in

our mouse colony. In the "Characterization of Nrg1 deletion by qPCR" section of the Methods, we describe the characterization of Nrg1 deletion in primary neuronal cultures from Nrg1 KO embryos and control littermates (Ctrl n = 4 and Nrg1 KO n = 4). Due to an oversight, we did not update Supplementary Figure 1 with this graph and inadvertently left data from an earlier experiment in which we tested the deletion in newborn pup littermates (Ctrl = 3 and Nrg1 KO = 2). We apologize for this error.

The correct graph has now been added to Supplementary Figure 1.

Reviewer #2

5. For all data sets please provide full statistical reports.

Reply:

We have included a supplemental table with the Statistical Report for each dataset, as suggested by Reviewer #2.

Reviewer #3 (Comments to the Authors (Required)):

General response to all Reviewers

We would like to thank all the three Reviewers for their effort and for their very positive and constructive feedback. We particularly appreciate that all three Reviewers agree that the evidence presented in the manuscript is sound and robustly supports the conclusions, indicating the value of our study to the community. For instance: Reviewer #1, "The data is clear and conclusions are supported by the experiments"; Reviewer #2, "Data presented in manuscript supports the conclusions drawn by authors"; Reviewer #3, "The study presents high-quality data and effectively presents its results".

As it is usually the case when multiple reviewers provide feedback on a manuscript, the feedback from the three Reviewers provided different perspectives and directions for investigation. As we understand it, the overall suggestions mainly encouraged us to provide further mechanistic insights into the role of Nrg1 in axonal development and to explore a possible role of Nrg1 at the adult stage.

We have significantly improved the manuscript by addressing the Reviewers' feedback. Specifically, to further improve the quality of our study, we i) identified GAP43 as a relevant effector of Nrg1 signaling, ii) deepened the analysis of the cellular effects of Nrg1 signaling in neurite outgrowth, and iii) explored the role of Nrg1 in maintaining callosal projections through loss-of-function experiments in the cortices of adult mice. In addition to these main points, we have responded extensively to the reviewers' input, as detailed point-by-point below.

Point-to-point rebuttal to Reviewer #3

Reviewer #3

The article titled "Nrg1 intracellular signaling regulates the development of interhemispheric callosal connections" explores the role of the schizophrenia (SZ) risk gene Nrg1 in the development of interhemispheric callosal connections. The authors demonstrate a cell-autonomous function of Nrg1 in excitatory neurons, where it influences axon growth in the corpus callosum. The findings reveal that Nrg1 deletion leads to underdeveloped axons, while overexpression of Nrg1 results in increased axon growth. This effect is attributed to the activation of Nrg1, leading to the release of the intracellular domain (ICD), which in turn promotes axon growth. The study presents high-quality data and effectively presents its results. However, certain concerns warrant attention:

Reply:

We extend our sincere appreciation to Reviewer #3 for the thorough assessment of our manuscript and the valuable insights provided. We are very pleased that the reviewer recognized our commitment to scientific quality and reliability. We address each of the points raised by the reviewer below.

Reviewer #3

1. The study highlights that Nrg1 knockout (KO) brains exhibit underdeveloped callosal axons and hypoconnectivity, yet there is no observed difference in the size of the corpus callosum. This observation contrasts with reports associating reduced corpus callosum size with SZ patients. It would be valuable to discuss how the hypoconnected corpus callosum maintains its size in Nrg1-null brains.

Reply:

In fact, we mention in our Introduction that a reduction in the size of the corpus callosum is one of the phenotypes reported in schizophrenic patients. To our understanding, the cellular basis for this phenotype in humans is not entirely clear, especially since schizophrenia is not typically characterized by neurodegeneration. It has been proposed that this reduction may be due to deficits in myelination, as discussed by Raabe et al, 2018 (PMID: 30451850).

In the case of the phenotype observed in Nrg1 KO mice, we believe that the most straightforward explanation is that Nrg1 deletion impairs axonal growth to an extent that results in a delay, but not a massive disruption, of callosal development.

To clarify this speculation, we have added the following sentence to the Results (page 6, line line 34):

“We did not find obvious differences in the thickness of Nrg1 KO callosal structure compared to control littermates nor other major histological abnormalities *suggesting that CC development is delayed but not severely disrupted in the absence of Nrg1* (Supplemental Fig. 1F).”

Reviewer #3

2. Dye tracking experiments were conducted on postnatal day 0 (P0) animals. It would be informative to extend the analysis to assess the state of these axons at postnatal day 30 (P30) to gain insight into the long-term effects of Nrg1 manipulation.

Reply:

We thank Reviewer #3 for this constructive suggestion. The long-term effects of Nrg1 manipulation on interhemispheric connections are indeed an interesting and valuable area for future investigation. As mentioned in our response to Reviewer #1, our current study focuses on the developmental role of Nrg1 in callosal axons. However, we agree that extending the analysis to later stages could enhance our understanding of Nrg1's role.

Dye tracking experiments in the adult brain present challenges due to differences in tissue properties that complicate dye diffusion. Nevertheless, to begin investigating Nrg1's role in interhemispheric connections at later stages, we performed gain- and loss-of-function experiments for analysis in adult stages. Here, we provide a summary of these experiments, as described in our response to Reviewer #1:

- Postnatal effects of Nrg1 expression: we performed in utero electroporation (IUE) experiments to assess the postnatal effects of Nrg1 expression. Unfortunately, we encountered significant challenges, in particular maternal infanticide due to perinatal stress, which limited the number of viable electroporated mice. This problem, which is unfortunately common in IUE, prevented us from obtaining sufficient data during the review period.

- Role of Nrg1 in adult callosal projection integrity: to assess whether Nrg1 signaling is essential for maintaining callosal projection integrity in adults, we used inducible UBC-CreER2 mice for targeted Nrg1 deletion. Our results, presented in Supplementary Figure 6, indicate no significant differences in the profile of callosal connections, suggesting that Nrg1 may play a redundant role once callosal projections are established. However, it remains plausible that Nrg1 influences synaptic plasticity or neurotransmitter release in callosal neurons at this stage. This would be consistent with previous research from our lab and others on the involvement of Nrg1 in cortical wiring and synaptic transmission.

Future studies, as suggested in the Discussion, could further explore these possibilities using synaptic and cellular markers to investigate Nrg1's contributions to interhemispheric synaptic wiring. We appreciate Reviewer #3's insightful suggestion and aim to address these aspects in future research efforts.

Reviewer

#3

3. Several studies have linked SZ with the overexpression of Nrg1, such as the work by Olaya et al., 2017, which reported Nrg1 type III overexpression in the brain leading to SZ-like behavior. Considering this, it is important to address the association between loss of function, axon undergrowth, and SZ. How does the observed axon undergrowth in the context of Nrg1 deletion relate to the reported Nrg1 overexpression and its behavioral consequences in SZ?

Reply:

We thank reviewer #3 for highlighting the interesting study by Olaya et al. from Cynthia Weickert's lab. Indeed, studies in humans and animal models have shown that both deficiency and excess of Nrg1 signaling can be detrimental to cortical development and function. Like many in the field, we wish we had a solid, evidence-based explanation for this observation. In our opinion, the most reasonable speculation is the one proposed by Agarwal and Schwab and others in 2014 (PMID: 25131210). Agarwal proposed a bell-shaped model in which an optimal amount of Nrg1 is required to maintain the correct balance of inhibitory/excitatory signaling to support proper neuronal development and wiring.

We have added a sentence to the Discussion to refer to these two studies and this working model for Nrg1 signaling (Discussion, page 12, line 1):

“Interestingly, while most studies in preclinical models have focused on loss of Nrg1/ErbB4 signaling, others have shown that exogenous expression of Nrg1 can also be detrimental to cortical wiring and lead to SZ-like symptoms (Hahn et al, 2006; Yin et al, 2013; Agarwal et al, 2014; Olaya et al, 2017). These results suggest that an optimal level of Nrg1 is required to maintain homeostasis of excitatory/inhibitory circuits in the cortex (Agarwal et al, 2014).”

Reviewer #3

4. Rescue experiments on Nrg1 KO cultures by transfecting with full-length Nrg1 and ICD versions, could provide insights into whether axon growth can be rescued to wild-type levels.

Reply:

We appreciate this valuable suggestion from Reviewer #3. As the reviewer suggests, performing rescue experiments to restore axonal growth in Nrg1 KO neurons is a logical next step to determine Nrg1 function. Based on this suggestion, we have designed experiments to directly address this point and further explore the mechanisms behind Nrg1 signaling in axonal growth. Specifically, we examined the effect of Nrg1 loss-of-function (Nes-Cre, deletion in primary cortical neurons) on major pathways involved in axonal growth, including AKT, JNK, ERK, and Growth Associated Protein 43 (GAP43). We were particularly excited to find that Nrg1 deletion led to a significant decrease in the expression of GAP43, a well-recognized player in axonal growth and regeneration (Chung et al., 2020; Tedeschi et al. 2009; Okada et al., 2022).

Given the importance of this target, we performed further experiments to support the involvement of GAP43 in Nrg1 signaling. Specifically, these experiments demonstrated that GAP43 expression could cell-autonomously rescue the loss of Nrg1 in primary cortical neurons in vitro. The results are shown in Figure 4 and Supplementary Figure 4.

This compelling finding not only enriches our understanding of Nrg1 signaling, but also sheds light on the intricate interplay between Nrg1 and GAP43 in the context of axonal growth. We trust that reviewer #3 will find these results both intriguing and valuable, underscoring our commitment to advancing mechanistic insights into Nrg1 signaling pathways.

Reviewer

#3

5. Given the conflicting reports regarding the role of Nrg1 in SZ, it would be valuable to explore the extent of Nrg1 expression in cells transfected with Nrg1 full-length and Nrg1-ICD variants. Additionally, investigating the degree of overexpression associated with axon overgrowth and its correlation with SZ-like behavior could provide a clearer understanding of the complex relationship between Nrg1 expression levels and SZ-associated phenotypes.

Reply:

We appreciate the reviewer's suggestion to investigate Nrg1 protein levels and their potential correlation with axon length. This is certainly an interesting avenue for future investigation.

However, as discussed in our response to Reviewer #2, variability in gene expression is an inherent feature of electroporation and transfection methods. While quantifying the correlation between Nrg1 expression and axon growth could be informative, it poses significant technical

challenges. Despite this limitation, we believe that our experimental design is robust and adequately supports the conclusions of the study.

June 11, 2024

RE: Life Science Alliance Manuscript #LSA-2023-02250-TR

Dr. Pietro Fazzari
Lab of Cortical Circuits in Health and Disease
Centro de Investigacion Principe Felipe, I64, c/ Eduardo Primo Yufero, 3
Valencia 46012
Spain

Dear Dr. Fazzari,

Thank you for submitting your revised manuscript entitled "Nrg1 intracellular signaling regulates the development of interhemispheric callosal connections". We would be happy to publish your paper in Life Science Alliance pending final revisions necessary to meet our formatting guidelines.

- please be sure that the authorship listing and order is correct
- please upload your main manuscript text as an editable doc file
- please add the Twitter handle of your host institute/organization as well as your own or/and one of the authors in our system
- titles in the system and manuscript file should match
- please consult our manuscript preparation guidelines <https://www.life-science-alliance.org/manuscript-prep> and make sure your manuscript sections are in the correct order
- please add your main, supplementary figure, and table legends to the main manuscript text after the references section

A. FINAL FILES:

B. MANUSCRIPT ORGANIZATION AND FORMATTING:

Sincerely,

REBUTTAL LETTER

Point-to-point Rebuttal letter for the Manuscript # LSA-2023-02250-TR – “*Nrg1 intracellular signaling regulates the development of interhemispheric callosal connections*” by Rodriguez-Prieto et al.

To facilitate the work of the Editors and Reviewers, we used the following formats to distinguish between Reviewer's comments, our response, and the text amended in the manuscript.

Formatting code:

Arial Blue, Reviewers feedback

Arial Black, our Response

Times new Roman Black, original the text of the Manuscript,

Times new Roman Green, changes to the original the text of the Manuscript

The position of the changes is referenced using the revised text (e.g. page 8, line 235).

Reviewer #1 (Comments to the Authors (Required)):

General response to all Reviewers

We would like to thank all the three Reviewers for their effort and for their very positive and constructive feedback. We particularly appreciate that all three Reviewers agree that the evidence presented in the manuscript is sound and robustly supports the conclusions, indicating the value of our study to the community. For instance: Reviewer #1, “The data is clear and conclusions are supported by the experiments”; Reviewer #2, “Data presented in manuscript supports the conclusions drawn by authors”; Reviewer #3, “The study presents high-quality data and effectively presents its results”.

As it is usually the case when multiple reviewers provide feedback on a manuscript, the feedback from the three Reviewers provided different perspectives and directions for investigation. As we understand it, the overall suggestions mainly encouraged us to provide further mechanistic insights into the role of Nrg1 in axonal development and to explore a possible role of Nrg1 at the adult stage.

We have significantly improved the manuscript by addressing the Reviewers' feedback. Specifically, to further improve the quality of our study, we i) identified GAP43 as a relevant effector of Nrg1 signaling, ii) deepened the analysis of the cellular effects of Nrg1 signaling in neurite outgrowth, and iii) explored the role of Nrg1 in maintaining callosal projections through loss-of-function experiments in the cortices of adult mice. In addition to these main points, we have responded extensively to the reviewers' input, as detailed point-by-point below.

Point-to-point rebuttal to Reviewer #1

Reviewer #1

The manuscript entitled “Nrg1 intracellular signaling regulates the development of interhemispheric callosal connections” (please add mention of the animal specie in title: “in mice”) by Angela Rodriguez-Prieto and collaborators explores the roles of Nrg1 in glutamatergic cortical neurons development. Specifically, the authors demonstrate that loss of Nrg1 leads to reduced axon length in vivo (through tracer injection) and in vitro, whereas Nrg1 overexpression increases axon length in vitro and in vivo (through in utero cortical electroporations). Interestingly, this effect can be observed as well by expressing the cleaved intracellular domain

of Nrg1, providing some mechanistic insight. The data is clear and conclusions are supported by the experiments. The advance from this paper is not major, since other papers demonstrate that Nrg1 affects axon length in gabaergic neurons (as discussed by the authors), but also in excitatory neurons (see for example Zhang et al. *Sci. Rep.* 2017, doi: 10.1038/srep42525). Still, this study has value to the community and I would support publication of a revised version. Below are suggestions that would reinforce the impact of the paper.

Reply:

We thank Reviewer # 1 for taking the time to evaluate our manuscript. We are particularly thankful for appreciating that our results are clear and support the conclusion of our study. We believe that this is particularly important in a complex study that presents both *in vivo* and *in vitro* approaches with mechanistic insights.

With regard to the novelty of the study, we thank the author for acknowledging that “this study has value to the community”. This feedback also stimulated us to improve the “Introduction” with regard to the novelty of our study in the context of the previous literature. Specifically:

Regarding the comment “add mention of the animal specie”, we amended the Title as requested.

New title: "Nrg1 intracellular signaling regulates the development of interhemispheric callosal connections in mice"

Concerning the comment, “Nrg1 affects axon length in gabaergic neurons (as discussed by the authors)”, we amended the text in the introduction to explain more clearly our previous findings on the role of Erbb4 activation inhibitory neurons. To facilitate the work of Reviewer #1 we paste here the amended paragraph that can be found in the revised Manuscript. In Green we highlight the amended Text.

See Introduction, page 4 line 5

“These studies demonstrated that Nrg1/Erbb4 signaling plays an important role in the cortex, and specifically in the wiring of inhibitory cortical neurons that express the Nrg1 receptor Erbb4 (Bjarnadottir et al, 2007; Li et al, 2007; Mei & Xiong, 2008; Chen et al, 2010a, 2010b; Fazzari et al, 2010, 2014; Pedrique & Fazzari, 2010; Rahman-Enyart et al, 2020; Navarro-Gonzalez et al, 2021). Erbb4 activation in inhibitory neurons is required for proper wiring of local inhibitory circuits, as it promotes the growth of inhibitory axons in vitro the formation of GABAergic synapses in vitro and in vivo (Fazzari et al, 2010; Rico & Marín, 2011; Navarro-Gonzalez et al, 2021).”

Regarding the role of Nrg1 in excitatory neurons, Reviewer #1 cited the manuscript of Zhang et al. Sci. Rep. 2017, doi: 10.1038/srep42525. We thank Reviewer #1 for mentioning this study. We did not include earlier this reference because we considered it redundant with Chen, Y.,... Talmage, D. A; *J. Neurosci.* 30, 9199-9208, which we have cited instead in both the Introduction and Discussion. The paper by Chen et al. in 2010 was, to our knowledge, the first to extensively investigate the role of Nrg1 in excitatory neurons. Moreover, we understand that Zhang et al. (2017) primarily focused on dendrites, based on their use of the dendritic marker MAP2 and the presented images. Notably, Zhang et al. [*Sci. Rep.* 2017, doi: 10.1038/srep42525] follow up on a previous manuscript from the same group, namely Zhang et al., *Sci Rep.* 2016. doi: 10.1038/srep19581. Zhang et al. 2016 present the analysis of neurite outgrowth in NRG1 KO primary neurons using the same approach (MAP2 labeling). Despite the fact that, to our understanding, Zhang et al. (2016 and 2017) focused on dendrites, for the sake of completeness we have now included their references in the amended manuscript.

See Discussion, page 12 line 8:

“The role of Nrg1 in axonal development in pyramidal neurons is poorly understood. Nonetheless, a few studies suggested that Nrg1 loss-of-function may impair dendritic development (Zhang et al, 2016, 2017).”

Reviewer #1:

1. On a mechanistic point of view, the authors suggest that Nrg1 cleavage and release of the ICD peptide is tied to the axon elongation phenotype (Figures 3 and 4). This could be further demonstrated by blocking Nrg1 cleavage either with a g-secretase inhibitor, or by expressing a Nrg1 mutant which cannot be cleaved (if this exists).

Reply:

This is a sensible suggestion. Indeed, previous studies have demonstrated that inhibiting gamma-secretase activity leads to a decrease in neurite outgrowth (see for instance Barao S. and De Strooper, 2016; Javier-Torrent et al., 2019; Nathalie Jurisch-Yaksi et al., 2013). While this finding supports our hypothesis regarding the involvement of Nrg1 cleavage in axonal development, caution must be exercised when employing gamma-secretase inhibitors as a tool. The primary concern lies in the broad substrate specificity of gamma-secretase, which processes over 50 proteins. Consequently, determining the optimal concentration of a gamma-secretase inhibitor may prove challenging; a low concentration might yield no discernible effect, while a high concentration could induce toxicity unrelated to Nrg1 processing, thereby impairing neuronal survival. Regarding the use of an uncleavable Nrg1, the Nrg1 V321L mutant reported by DeJaegere et al. (2008) would be a possible option. However, this mutation only partially reduces Nrg1 processing, making this option less straightforward.

In summary, while the reviewer's suggestion is quite reasonable, its implementation is not straightforward due to the variety of potential gamma-secretase targets implicated in neurite growth and survival, in addition to Nrg1. Overall, we believe that the expression of Nrg1-ICD represents the most specific experiment to elucidate the role of Nrg1 intracellular signaling, as demonstrated by our work and that of others (Fazzari, 2014; Navarro-González, 2019; Bao, 2003).

Reviewer

#1:

2. Quantification of axon length could be completed with quantifications of axon ramification (number of branchpoints) as well as dendritic morphology to describe if the phenotype is axon specific or if the whole neuronal morphology is affected.

Reply:

We appreciate the reviewer's suggestion to further strengthen the cellular analysis of the role of Nrg1 signaling in neurite outgrowth. In response to reviewer #1's constructive input, we have expanded our study to include quantification of axonal branching patterns (number of branch points per length) in all *in vitro* experiments. The additional data are shown in the Main Figures. We also performed a Sholl analysis of dendritic arborization in all experimental conditions. The quantifications of the Sholl analysis are shown in Supplementary Figures.

Reviewer

#1:

3. The observation of a growth phenotype at P2 is intriguing, but raises the question if accelerated growth has some consequences on cortical circuits building. Would axons growing faster continue to develop abnormal connections? Reach the wrong targets? or rather, would they prune because they reach targets before the local environment is favourable? Looking at a later timepoint (eg. P30) where axonal connections are fully developed would provide much information.

Reply:

Reviewer #1 raises an interesting point, which is certainly worthy of future investigation. Callosal development is a complex, multi-step process. As the reviewer points out, our observation that Nrg1 expression by IUE accelerates callosal axon growth doesn't necessarily translate into improved interhemispheric connectivity at later stages. We agree that Nrg1-induced accelerated growth may be compensated for later by pruning and target refinement. Alternatively, Nrg1-expressing neurons might overshoot and reach inappropriate targets.

Our current study focuses primarily on the early stages of callosal development and does not address the role of Nrg1 in later stages of interhemispheric connectivity. However, we appreciate the suggestion to explore the function of Nrg1 at later stages, which could enrich our understanding of callosal circuitry. Therefore, we undertook the following experimental approaches to address this suggestion:

- First, we performed IUE experiments to assess the postnatal effects of Nrg1 expression. Unfortunately, we were unable to obtain sufficient numbers of electroporated mice due to maternal infanticide in mothers experiencing perinatal stress. This is a recognized problem in this experimental paradigm and we were unable to overcome it during the revision process.

- Second, we investigated whether Nrg1 signaling is necessary for maintaining callosal projection integrity in adults by using inducible UBC-CreER2 mice for targeted Nrg1 deletion. Our findings, presented in Supplementary Figure 6, indicate no significant differences in the profile of the callosal connections, suggesting that Nrg1 may play a redundant role once callosal projections are established. However, it remains possible that Nrg1 influences synaptic plasticity or neurotransmitter release in callosal neurons at this stage. This hypothesis would be consistent with previous studies from our lab and others regarding Nrg1's role in cortical wiring and synaptic transmission (see references in the manuscript).

Future studies, as suggested in the manuscript (page 9, line 27), could explore these possibilities using synaptic and cellular markers to investigate the contributions of Nrg1 to interhemispheric synaptic wiring.

Reviewer #1

Technical comments

1. Please provide graphical representation that show individual datapoints so that the reader can appreciate experimental variability. Histograms and error bars represent a loss of information for the reader.

Reply:

We modified all the graphs to include the individual datapoints as suggested.

Reviewer #1

2. Figure 1: It is hard to visualize axon development from the highly truncated pictures, especially since there is no image showing brain slice morphology (aside from a cartoon). This information is important for the reader to demonstrate that images are taken at the same brain level (as should be for stereotactic injections) and that the orientation of the cut is perfectly transversal. Any shift in the rostro-caudal axis or laterally could provide an effect mimicking changes in axon growth. Similarly, since on average 4-5 slices were collected per brain, what is the variability within one single animal? What are the differences of position of the slice on a rostro-caudal axis? One might advise to show and quantify only one slice (always the same position) per animal, if there's a robust effect of reduction on axon length then it should be evident as well.

Reply:

We appreciate the critical input from reviewer #1. Indeed, the *in vivo* tracing experiments shown in Figures 1 and 4 are complex and require precise execution. We were fortunate to

collaborate with two highly specialized laboratories: the lab of Guillermina Lopez-Bendito for the dye tracing in Figure 1 (e.g. Moreno-Juan et al., Nat Comm, 2017; Antón-Bolaños N. et al., Science, 2019) and the lab of Cristina Gil-Sanz for in utero electroporation (e.g. Fabra-Beser J et al., J Neuro, 2021; Gil-Sanz C. et al., Neuron, 2015). We took great care with technical aspects such as brain orientation and included only consistently labeled and processed specimens in our analysis.

In response to the reviewer's suggestions, we have improved the quality of Figure 1. We have selected new representative images for control and Nrg1 KO mice and included DAPI staining (Figure 1). In addition, we have added a low magnification image in the Supplementary Material to show the entire section as suggested (Supplemental Figure 1D).

Regarding the number of sections and the suggestion to quantify only one section, we thank reviewer #1 for prompting us to clarify our methodology. We sectioned and photographed the entire brain, but performed the analysis at a specific and limited rostrocaudal level in the somatosensory cortex. We used the Allen Brain Atlas at P1, Nissl, coronal sections, section #110 (<https://developingmouse.brain-map.org/static/atlas>) as a reference, which we have now added to the Materials and Methods section. Because we labeled both sides of the brain with different dyes (as described in the Methods section), we obtained two measurements per section, which increases the robustness of our analysis. Although rostro-caudal diffusion of the dye may vary, this approach allowed us to obtain an average of six measurements from three consecutive sections. Each section was 80 μm thick, covering a total of 240 μm , a relatively small span, resulting in very consistent data.

We believe these revisions effectively address Reviewer #1's concerns and significantly improve the clarity and strength of Figure 1.

Reviewer #1

3. Figure 4: similarly to Figure 1, it would be better for the reader to present a picture of the entire slice and to explain if the sections chosen for the quantification have been taken at the same position on a rostro-caudal axis. Once again, changes in the cutting plan can have artificial impact on the quantification.

Reply:

Similar to the previous point, axonal growth was consistently quantified at the rostro-caudal level of the somatosensory cortex using the reference Allen Brain Atlas at P1, Nissl, coronal sections, section #110 (<https://developingmouse.brain-map.org/static/atlas>). We focused on this region because, at this rostro-caudal level, callosal axons grow parallel to the orientation of the section at this stage. Therefore, quantification of axonal elongation at this rostro-caudal level is straightforward and robust. We have also included a low magnification image of the sections with DAPI counterstaining in Supplementary Figure 5C,D as suggested.

Reviewer #1

4. Quantifications of axon length in vitro from figures 2 and 3 should be presented as absolute values in μm rather than normalized values. If normalization was rendered necessary because of culture to culture variability, the raw data can be presented in supplementary. Still this information is important to the reader to compare to the literature and estimate the viability of neuronal cultures. Specifically, from the scale bar and the pictures and traces in figure 3, one might think that Ctrl neurons in figure 3 are actually less developed than the KO neurons in figure 2. If so, one might overestimate the magnitude of the axon elongation phenotype upon overexpression of Nrg1 and Nrg1-ICD.

Reply:

We welcome the opportunity to provide further clarification of our methodology. We agree that presenting axon length quantifications in absolute values (μm) may provide additional context for the reader to compare with the existing literature. To address this concern, we have

included the absolute raw values of axon length in Supplementary Figures (FS2, FS3, FS4). Statistical analysis of the non-normalized raw data strongly confirms the significance of the differences between experimental conditions and controls, further strengthening the robustness of our results.

We have left the evaluation of fold change relative to internal controls in the main figure because we are convinced that this is a well-established and scientifically sound approach in the field to compensate for normal experimental variability. In this regard, we respectfully invite Reviewer #1 to review our previous publications and those of other reputable labs (such as the labs of Oscar Marin, Beatriz Rico, Bart De Strooper, and Carlos Dotti; see the references in the manuscript for details).

Regarding the observed variability in neuronal development between the control conditions in Figures 2 and 3, it is well within the expected range for this experimental model and does not affect the overall conclusions of our study. Certainly, this small difference is not due to cell viability issues. Indeed, regarding the issue of "the viability of neuronal cultures", we perform stringent quality control measures, including thorough microscopic evaluation on different days of cell density, morphology, and absence of cell debris, to ensure the health and viability of our neuronal cultures. Any cultures showing signs of compromised viability or abnormal development are discarded.

In summary, we have addressed the reviewer's concern by including the absolute values of axon length in microns in the Supplementary Figures. We have also clarified our quality control measures and the rationale for the use of internal controls in this experimental paradigm. Overall, we believe that these changes strengthen the manuscript and provide the reader with a more complete understanding of our findings.

Reviewer #1

5. Figure 4 paned D: it seems that axons are fragmented in both the control and ICD conditions. Is this an artefact due to tissue fixation/preparation? If so, please provide a better picture. Otherwise, it could be that on this batch of electroporation there has been some sort of neurotoxic effect of plasmid expression, which could affect axon growth and decreases confidence in the result.

Reply:

We appreciate Reviewer #1's careful examination of Figure 4D. In our experience working with *in utero electroporation*, which is the expertise of Cristina Gil-Sanz's lab, it is not uncommon to observe some varicosities or irregularities in dendrites and axons at high magnification in these *in vivo* preparations (see for instance Guo et al., Nat Commun 2015, PMID: 26206566; and Fabra-Beser et la., J Neuro 2021, PMID: 34266896). Importantly, these irregularities are seen in 1) top-growing axons, suggesting that they are not related to impaired growth and 2) in both GFP control and Nrg1-expressing neurons, arguing against a neurotoxic effect of Nrg1 expression.

Regarding the quality of the images, while we could enhance the images by saturating them for more visually pleasing results, we prefer to maintain the integrity of the data by presenting images that are as close as possible to the raw data. Therefore, with the kind permission of Reviewer #1 and the editor, we would prefer to keep the images as they are.

Reviewer #2 (Comments to the Authors (Required)):

General response to all Reviewers

We would like to thank all the three Reviewers for their effort and for their very positive and constructive feedback. We particularly appreciate that all three Reviewers agree that the evidence presented in the manuscript is sound and robustly supports the conclusions, indicating the value of our study to the community. For instance: Reviewer #1, “The data is clear and conclusions are supported by the experiments”; Reviewer #2, “Data presented in manuscript supports the conclusions drawn by authors”; Reviewer #3, “The study presents high-quality data and effectively presents its results”.

As it is usually the case when multiple reviewers provide feedback on a manuscript, the feedback from the three Reviewers provided different perspectives and directions for investigation. As we understand it, the overall suggestions mainly encouraged us to provide further mechanistic insights into the role of Nrg1 in axonal development and to explore a possible role of Nrg1 at the adult stage.

We have significantly improved the manuscript by addressing the Reviewers' feedback. Specifically, to further improve the quality of our study, we i) identified GAP43 as a relevant effector of Nrg1 signaling, ii) deepened the analysis of the cellular effects of Nrg1 signaling in neurite outgrowth, and iii) explored the role of Nrg1 in maintaining callosal projections through loss-of-function experiments in the cortices of adult mice. In addition to these main points, we have responded extensively to the reviewers' input, as detailed point-by-point below.

Point-to-point rebuttal to Reviewer #2

The major risk factor for schizophrenia, Neuregulin 1 (NRG1), plays a crucial role in myelination, neurite growth, and spine formation. Here, the authors investigated the influence of NRG1 on callosal axon length and long-range cortico-cortical connectivity in relation to schizophrenia. The study reveals that Nrg1 deficiency affects callosal axon development in the mouse brain. Similarly, Nrg1-deficient primary neurons exhibit shortened axons. Conversely, gain-of-function experiments using Nrg1-FL and Nrg1-ICD overexpression clearly display heightened axon length in cultured neurons and in the brain. Data presented in manuscript supports the conclusions drawn by authors. While there is potentially new and useful information here and the results appear interesting, the authors presentation lacks details and additional evaluations to validate their experimental models. The presentation of the figures could be better, and the authors need to discuss functional and behavioral aspects of the data and its interpretation. A few concerns exist:

Reply:

We thank Reviewer #2 for all the valuable feedback on our manuscript. We appreciate the acknowledgement of the soundness of our findings on the role of Neuregulin 1 (NRG1) and the conclusions we draw from these results. We have improved our discussion to clarify the functional and behavioral significance of NRG1 in neuronal development and schizophrenia. In addition, we have significantly improved the presentation of the figures and further clarified the methodology, including the statistical methodology. We believe that this revision fully addresses the comments of Reviewer #2.

Reviewer #2

Major:

1. Schizophrenic patients exhibit cognitive impairments and alter social behavior. The behavioral phenotype of Nrg1-deficient mice should be discussed in this context.

Reply:

We agree with the reviewer that the behavioral impact of Nrg1 deficiency is relevant. We have revised the text to acknowledge the reviewer's point and provided references to the relevant literature. While our current focus is on cellular and molecular neurobiology, we have

included a reference to key studies demonstrating SZ-like behavioral phenotypes in mice mutant for Nrg1/ErbB4 forward and intracellular signaling (page 3, line 85). These studies, extensively reviewed by Mei (2008) and Mei & Xiong (2014), provide strong evidence for the link between Nrg1 signaling and schizophrenia-related behaviors.

Page 3, line 30:

“Numerous studies have identified the *neuregulin 1 (NRG1)* gene as a risk factor for the development of schizophrenia in various populations (Stefansson et al, 2002; Williams et al, 2003; Yang et al, 2003; Shyu et al, 2004; Tang et al, 2004; Harrison & Weinberger, 2005; Mei & Xiong, 2008). Interestingly, several studies in preclinical mouse models have shown that various genetic mutations that impair Nrg1/ErbB4 forward and intracellular signaling exhibit SZ-like symptoms, such as working memory deficits and hypersensitivity to psychostimulants (Stefansson et al, 2002; Coolen et al, 2005; Dejaegere et al, 2008; Mei & Xiong, 2008; Mei & Nave, 2014).”

Reviewer #2

2. Gender-based differences have been reported in patients with schizophrenia on onset, symptoms, and social behavior. The present study does not clearly state the exact number or ratio of male and female mice used. What was the rationale for such a selection? And were there any differences in results between male and female mice?

Reply:

We appreciate Reviewer #2 raising the important issue of sex differences in schizophrenia. However, our study focused on primary neuronal cultures and neonatal pups. In these experimental paradigms, sexual dimorphism is not well established in mice. To our knowledge, the current scientific literature does not address sex differences in mouse callosal development at these early developmental stages.

For the neuronal cultures, our model uses single-cell transfection, and these cultures are typically established by pooling cortical tissue from multiple embryos without regard to sex. Thus, the potential impact of sex differences is minimized in this context.

We hope this clarifies the rationale behind our experimental design and addresses the reviewer's concern.

Reviewer

#2

3. The functional aspect of the connectivity deficit in relation to the pathophysiology of schizophrenia should be discussed in detail.

Reply:

We appreciate the reviewer's suggestion to discuss the functional aspect of the altered connectivity observed in Nrg1-deficient mice and its potential implications in schizophrenia. In response, we have revised the Discussion section to more clearly articulate how the alterations in corpus callosum development observed in Nrg1-deficient mice may affect interhemispheric connectivity and brain function.

Specifically, in the original manuscript, we devoted four paragraphs of the Discussion (from page 10, line 14 to page 11, line 19) to describing the development of the corpus callosum and its pathophysiological functions in humans. In response to the reviewer's feedback, we have modified this section to more clearly describe how the alterations in corpus callosum development observed in Nrg1-deficient mice may affect interhemispheric connectivity and brain function (Discussion, page 10, line 14 to page 11, line 19).

We believe that these revisions significantly improve the clarity and readability of the Discussion and fully address the reviewer's feedback. We thank the reviewer for the valuable input.

Reviewer #2

4. For the characterization of *in vivo* NRG1 deletion, authors should confirm change at the protein level using staining, blotting, or similar techniques. Immunostaining data could be useful to understand region- and cell-specific changes in NRG1 levels in the brain.

Reply:

We thank the reviewer for this suggestion. We would like to emphasize that the Nrg1 flox mouse model is well established and numerous studies over the past 20 years have confirmed that Nrg1 is effectively deleted upon Cre expression (J:69623 Yang X, et al., Patterning of muscle acetylcholine receptor gene expression in the absence of motor innervation. *Neuron*. 2001 May;30(2):399-410 was the first reference. 20 references can be found at <https://www.informatics.jax.org/reference/allele/MGI:2447761?typeFilter=Literature>)

In our study, we further validated this deletion by showing that Nrg1 mRNA expression is abolished using qPCR. Since protein expression is not possible in the absence of mRNA, we believe that our approach provides an accurate and sensitive confirmation of Nrg1 deletion.

In addition, commercially available antibodies against NRG1 are not very reliable. Therefore, we respectfully suggest that a Western blot would not significantly increase the validity of our results compared to the qPCR data we have provided.

Reviewer #2

5. Similar to above, what is the NRG1 protein level in Nrg1 KO and Nrg1-FL/Nrg1-ICD overexpressing neurons compared to control neurons? Western blot data will be useful in this context.

Reviewer #2

6. Since overexpression of Nrg1-FL/Nrg1-ICD not only restores axon length but can also increase axon length more than that of control neurons, Is there any correlation between NRG1 protein level and axon length in cultured neurons?

Reply to points 5 and 6:

We reply together to the reviewer comments 5 and 6 since they are related.

We appreciate the reviewer's valuable suggestions regarding directly measuring Nrg1 protein levels and their potential correlation with axon length.

Our study design utilizes co-culturing with non-electroporated neurons to achieve sparse labeling and single-cell resolution. This approach offers the significant advantage of analyzing Nrg1 manipulation effects at the single-cell level while maintaining an environment with endogenous Nrg1 levels. However, it presents a challenge for Western blot analysis, as it wouldn't definitively distinguish between exogenous and endogenous Nrg1.

As the reviewer rightly pointed out, electroporation inherently leads to variable Nrg1 expression within the cell population, making quantification of this heterogeneity within the entire pool a significant challenge. While a theoretical correlation between Nrg1 levels and axon length may exist, our current quantification methods represent the average for the Nrg1-expressing population, which is a standard caveat of electroporation and transfection experiments.

In summary, while we appreciate the reviewer's suggestions, we acknowledge the technical challenges associated with the proposed Western blot analysis and correlation quantification. We believe that the current data, demonstrating the effects of Nrg1 manipulation on axon length, remains informative and sufficient to support the study's conclusions.

Reviewer #2

7. The adverse effects of NRG1 overexpression should also be added to the discussion.

Reply:

We agree with reviewer #2 and appreciate this constructive suggestion. In response, we have modified the Discussion to include relevant references on this topic and its relevance to inhibitory/excitatory homeostasis in the brain. Specifically, we have added the following paragraph to the Discussion (page 12, line 1):

“Interestingly, while most studies in preclinical models have focused on loss of Nrg1/ErbB4 signaling, others have shown that exogenous expression of Nrg1 can also be detrimental to cortical wiring and lead to SZ-like symptoms (Hahn et al, 2006; Yin et al, 2013; Agarwal et al, 2014; Olaya et al, 2017). These results suggest that an optimal level of Nrg1 is required to maintain homeostasis of excitatory/inhibitory circuits in the cortex (Agarwal et al, 2014).”

Reviewer

#2

Minor:

1. Graphs should show individual data points and bidirectional error bar.

Reply:

The graphs were modified according to the reviewer's suggestion.

Reviewer #2

2. What is the rational of using relative ratio in figure 2B, 3C, 4B and 4E rather than raw reading?

Reply:

We thank Reviewer #2 for this feedback. As mentioned in detail in our response to Reviewer #1 (Technical comment 4), using an internal control as a reference to evaluate the effect of gene expression or deletion is a well-established and commonly used approach, particularly in early neurodevelopmental studies. This method addresses inherent experimental variability and biological noise, especially when working with primary neuronal cultures. Normalizing the data to an internal control enhances the consistency and reliability of the analysis. Additionally, in response to Reviewer #1's specific suggestion, we have included the absolute raw values of axon length for all primary neuronal culture experiments in the Supplementary Figures (SF2, SF3, SF4). This provides additional context for readers who prefer raw data.

Reviewer #2

3. Please describe the method used to identify outliers in data?

Reply:

We thank Reviewer #2 for raising this issue. We used the iterative Grubbs test with an alpha of 0.01 in GraphPad Prism to identify outliers. This test worked well for our data set and demonstrated robustness and consistency in our analysis. We have added the use of the iterative Grubbs test with the alpha value to the Methodology section (page 22, line1). In addition, the results of this outlier analysis are described in the statistical report table as suggested in item 5.

Reviewer #2

4. In figure legend of supplementary figure 1B. it mentioned "Ctrl = 3 and Nrg1 KO = 2, littermates"; please explain the n number. Did authors used only two Nrg1 KO pups to obtain the data in supplementary Figure 1B and draw conclusions?

6. The method section describes mRNA isolation from cultured neurons, but there are no related data.

Reply to points 4 and 6:

We thank reviewer #2 for pointing out these issues. We acknowledge an editing error in Supplementary Figure 1B and apologize for the confusion.

The nestin Cre-deleter drives Cre expression as early as embryonic day 11. This line has been validated in many publications and we have carefully verified that it works as expected in our mouse colony. In the "Characterization of Nrg1 deletion by qPCR" section of the Methods, we describe the characterization of Nrg1 deletion in primary neuronal cultures from Nrg1 KO embryos and control littermates (Ctrl n = 4 and Nrg1 KO n = 4). Due to an oversight, we did not update Supplementary Figure 1 with this graph and inadvertently left data from an earlier experiment in which we tested the deletion in newborn pup littermates (Ctrl = 3 and Nrg1 KO = 2). We apologize for this error.

The correct graph has now been added to Supplementary Figure 1.

Reviewer #2

5. For all data sets please provide full statistical reports.

Reply:

We have included a supplemental table with the Statistical Report for each dataset, as suggested by Reviewer #2.

Reviewer #3 (Comments to the Authors (Required)):

General response to all Reviewers

We would like to thank all the three Reviewers for their effort and for their very positive and constructive feedback. We particularly appreciate that all three Reviewers agree that the evidence presented in the manuscript is sound and robustly supports the conclusions, indicating the value of our study to the community. For instance: Reviewer #1, "The data is clear and conclusions are supported by the experiments"; Reviewer #2, "Data presented in manuscript supports the conclusions drawn by authors"; Reviewer #3, "The study presents high-quality data and effectively presents its results".

As it is usually the case when multiple reviewers provide feedback on a manuscript, the feedback from the three Reviewers provided different perspectives and directions for investigation. As we understand it, the overall suggestions mainly encouraged us to provide further mechanistic insights into the role of Nrg1 in axonal development and to explore a possible role of Nrg1 at the adult stage.

We have significantly improved the manuscript by addressing the Reviewers' feedback. Specifically, to further improve the quality of our study, we i) identified GAP43 as a relevant effector of Nrg1 signaling, ii) deepened the analysis of the cellular effects of Nrg1 signaling in neurite outgrowth, and iii) explored the role of Nrg1 in maintaining callosal projections through loss-of-function experiments in the cortices of adult mice. In addition to these main points, we have responded extensively to the reviewers' input, as detailed point-by-point below.

Point-to-point rebuttal to Reviewer #3

Reviewer #3

The article titled "Nrg1 intracellular signaling regulates the development of interhemispheric callosal connections" explores the role of the schizophrenia (SZ) risk gene Nrg1 in the development of interhemispheric callosal connections. The authors demonstrate a cell-autonomous function of Nrg1 in excitatory neurons, where it influences axon growth in the corpus callosum. The findings reveal that Nrg1 deletion leads to underdeveloped axons, while overexpression of Nrg1 results in increased axon growth. This effect is attributed to the activation of Nrg1, leading to the release of the intracellular domain (ICD), which in turn promotes axon growth. The study presents high-quality data and effectively presents its results. However, certain concerns warrant attention:

Reply:

We extend our sincere appreciation to Reviewer #3 for the thorough assessment of our manuscript and the valuable insights provided. We are very pleased that the reviewer recognized our commitment to scientific quality and reliability. We address each of the points raised by the reviewer below.

Reviewer #3

1. The study highlights that Nrg1 knockout (KO) brains exhibit underdeveloped callosal axons and hypoconnectivity, yet there is no observed difference in the size of the corpus callosum. This observation contrasts with reports associating reduced corpus callosum size with SZ patients. It would be valuable to discuss how the hypoconnected corpus callosum maintains its size in Nrg1-null brains.

Reply:

In fact, we mention in our Introduction that a reduction in the size of the corpus callosum is one of the phenotypes reported in schizophrenic patients. To our understanding, the cellular basis for this phenotype in humans is not entirely clear, especially since schizophrenia is not typically characterized by neurodegeneration. It has been proposed that this reduction may be due to deficits in myelination, as discussed by Raabe et al, 2018 (PMID: 30451850).

In the case of the phenotype observed in Nrg1 KO mice, we believe that the most straightforward explanation is that Nrg1 deletion impairs axonal growth to an extent that results in a delay, but not a massive disruption, of callosal development.

To clarify this speculation, we have added the following sentence to the Results (page 6, line line 34):

“We did not find obvious differences in the thickness of Nrg1 KO callosal structure compared to control littermates nor other major histological abnormalities suggesting that CC development is delayed but not severely disrupted in the absence of Nrg1 (Supplemental Fig. 1F).”

Reviewer #3

2. Dye tracking experiments were conducted on postnatal day 0 (P0) animals. It would be informative to extend the analysis to assess the state of these axons at postnatal day 30 (P30) to gain insight into the long-term effects of Nrg1 manipulation.

Reply:

We thank Reviewer #3 for this constructive suggestion. The long-term effects of Nrg1 manipulation on interhemispheric connections are indeed an interesting and valuable area for future investigation. As mentioned in our response to Reviewer #1, our current study focuses on the developmental role of Nrg1 in callosal axons. However, we agree that extending the analysis to later stages could enhance our understanding of Nrg1's role.

Dye tracking experiments in the adult brain present challenges due to differences in tissue properties that complicate dye diffusion. Nevertheless, to begin investigating Nrg1's role in interhemispheric connections at later stages, we performed gain- and loss-of-function experiments for analysis in adult stages. Here, we provide a summary of these experiments, as described in our response to Reviewer #1:

- Postnatal effects of Nrg1 expression: we performed in utero electroporation (IUE) experiments to assess the postnatal effects of Nrg1 expression. Unfortunately, we encountered significant challenges, in particular maternal infanticide due to perinatal stress, which limited the number of viable electroporated mice. This problem, which is unfortunately common in IUE, prevented us from obtaining sufficient data during the review period.

- Role of Nrg1 in adult callosal projection integrity: to assess whether Nrg1 signaling is essential for maintaining callosal projection integrity in adults, we used inducible UBC-CreER2 mice for targeted Nrg1 deletion. Our results, presented in Supplementary Figure 6, indicate no significant differences in the profile of callosal connections, suggesting that Nrg1 may play a redundant role once callosal projections are established. However, it remains plausible that Nrg1 influences synaptic plasticity or neurotransmitter release in callosal neurons at this stage. This would be consistent with previous research from our lab and others on the involvement of Nrg1 in cortical wiring and synaptic transmission.

Future studies, as suggested in the Discussion, could further explore these possibilities using synaptic and cellular markers to investigate Nrg1's contributions to interhemispheric synaptic wiring. We appreciate Reviewer #3's insightful suggestion and aim to address these aspects in future research efforts.

Reviewer

#3

3. Several studies have linked SZ with the overexpression of Nrg1, such as the work by Olaya et al., 2017, which reported Nrg1 type III overexpression in the brain leading to SZ-like behavior. Considering this, it is important to address the association between loss of function, axon

undergrowth, and SZ. How does the observed axon undergrowth in the context of Nrg1 deletion relate to the reported Nrg1 overexpression and its behavioral consequences in SZ?

Reply:

We thank reviewer #3 for highlighting the interesting study by Olaya et al. from Cynthia Weickert's lab. Indeed, studies in humans and animal models have shown that both deficiency and excess of Nrg1 signaling can be detrimental to cortical development and function. Like many in the field, we wish we had a solid, evidence-based explanation for this observation. In our opinion, the most reasonable speculation is the one proposed by Agarwal and Schwab and others in 2014 (PMID: 25131210). Agarwal proposed a bell-shaped model in which an optimal amount of Nrg1 is required to maintain the correct balance of inhibitory/excitatory signaling to support proper neuronal development and wiring.

We have added a sentence to the Discussion to refer to these two studies and this working model for Nrg1 signaling (Discussion, page 12, line 1):

“Interestingly, while most studies in preclinical models have focused on loss of Nrg1/ErbB4 signaling, others have shown that exogenous expression of Nrg1 can also be detrimental to cortical wiring and lead to SZ-like symptoms (Hahn et al, 2006; Yin et al, 2013; Agarwal et al, 2014; Olaya et al, 2017). These results suggest that an optimal level of Nrg1 is required to maintain homeostasis of excitatory/inhibitory circuits in the cortex (Agarwal et al, 2014).”

Reviewer #3

4. Rescue experiments on Nrg1 KO cultures by transfecting with full-length Nrg1 and ICD versions, could provide insights into whether axon growth can be rescued to wild-type levels.

Reply:

We appreciate this valuable suggestion from Reviewer #3. As the reviewer suggests, performing rescue experiments to restore axonal growth in Nrg1 KO neurons is a logical next step to determine Nrg1 function. Based on this suggestion, we have designed experiments to directly address this point and further explore the mechanisms behind Nrg1 signaling in axonal growth. Specifically, we examined the effect of Nrg1 loss-of-function (Nes-Cre, deletion in primary cortical neurons) on major pathways involved in axonal growth, including AKT, JNK, ERK, and Growth Associated Protein 43 (GAP43). We were particularly excited to find that Nrg1 deletion led to a significant decrease in the expression of GAP43, a well-recognized player in axonal growth and regeneration (Chung et al., 2020; Tedeschi et al. 2009; Okada et al., 2022).

Given the importance of this target, we performed further experiments to support the involvement of GAP43 in Nrg1 signaling. Specifically, these experiments demonstrated that GAP43 expression could cell-autonomously rescue the loss of Nrg1 in primary cortical neurons in vitro. The results are shown in Figure 4 and Supplementary Figure 4.

This compelling finding not only enriches our understanding of Nrg1 signaling, but also sheds light on the intricate interplay between Nrg1 and GAP43 in the context of axonal growth. We trust that reviewer #3 will find these results both intriguing and valuable, underscoring our commitment to advancing mechanistic insights into Nrg1 signaling pathways.

Reviewer

#3

5. Given the conflicting reports regarding the role of Nrg1 in SZ, it would be valuable to explore the extent of Nrg1 expression in cells transfected with Nrg1 full-length and Nrg1-ICD variants. Additionally, investigating the degree of overexpression associated with axon overgrowth and its correlation with SZ-like behavior could provide a clearer understanding of the complex relationship between Nrg1 expression levels and SZ-associated phenotypes.

Reply:

We appreciate the reviewer's suggestion to investigate Nrg1 protein levels and their potential correlation with axon length. This is certainly an interesting avenue for future investigation.

However, as discussed in our response to Reviewer #2, variability in gene expression is an inherent feature of electroporation and transfection methods. While quantifying the correlation between Nrg1 expression and axon growth could be informative, it poses significant technical challenges. Despite this limitation, we believe that our experimental design is robust and adequately supports the conclusions of the study.

June 14, 2024

RE: Life Science Alliance Manuscript #LSA-2023-02250-TRR

Dr. Pietro Fazzari
Centro de Investigacion Principe Felipe
Lab of Cortical Circuits in Health and Disease
Centro de Investigacion Principe Felipe, I64, c/ Eduardo Primo Yufera, 3
Valencia 46012
Spain

Dear Dr. Fazzari,

Thank you for submitting your Research Article entitled "Nrg1 intracellular signaling regulates the development of interhemispheric callosal axons in mice". It is a pleasure to let you know that your manuscript is now accepted for publication in Life Science Alliance. Congratulations on this interesting work.

DISTRIBUTION OF MATERIALS:

Again, congratulations on a very nice paper. I hope you found the review process to be constructive and are pleased with how the manuscript was handled editorially. We look forward to future exciting submissions from your lab.

Sincerely,
